# Influence of NAFLD and bariatric surgery on hepatic and adipose tissue mitochondrial biogenesis and respiration

Julie S. Pedersen [1✉], Marte O. Rygg[1], Karoline Chrøis[2], Elahu G. Sustarsic [3], Zach Gerhart-Hines [3], Nicolai J. Wever Albrechtsen [4,5,6], Reza R. Serizawa[7], Viggo B. Kristiansen[8], Astrid L. Basse[3], Astrid E. B. Boilesen[8], Beth H. Olsen[9], Torben Hansen [3], Lise Lotte Gluud[1], Sten Madsbad[10], Steen Larsen[2,11,14], Flemming Bendtsen[1,12,14] & Flemming Dela [2,13,14]

Impaired mitochondrial oxidative phosphorylation (OXPHOS) in liver tissue has been hypothesised to contribute to the development of nonalcoholic steatohepatitis in patients with nonalcoholic fatty liver disease (NAFLD). It is unknown whether OXPHOS capacities in human visceral (VAT) and subcutaneous adipose tissue (SAT) associate with NAFLD severity and how hepatic OXPHOS responds to improvement in NAFLD. In biopsies sampled from 62 patients with obesity undergoing bariatric surgery and nine control subjects without obesity we demonstrate that OXPHOS is reduced in VAT and SAT while increased in the liver in patients with obesity when compared with control subjects without obesity, but this was independent of NAFLD severity. In repeat liver biopsy sampling in 21 patients with obesity 12 months after bariatric surgery we found increased hepatic OXPHOS capacity and mitochondrial DNA/nuclear DNA content compared with baseline. In this work we show that obesity has an opposing association with mitochondrial respiration in adipose- and liver tissue with no overall association with NAFLD severity, however, bariatric surgery increases hepatic OXPHOS and mitochondrial biogenesis.

[1] Gastrounit, Medical Division, Copenhagen University Hospital Hvidovre, Kettegaard Allé 30, 2650 Hvidovre, Denmark. [2] Department of Biomedical Sciences, Center for Healthy Ageing, Faculty of Health and Medical Sciences, University of Copenhagen, Blegdamsvej 3B, 2200 Copenhagen, Denmark. [3] NNF Center for Basic Metabolic Research, Faculty of Health and Medical Sciences, University of Copenhagen, Blegdamsvej 3B, 2200 Copenhagen, Denmark. [4] Department of Biomedical Sciences, Faculty of Health and Medical Sciences, University of Copenhagen, Blegdamsvej 3B, 2200 Copenhagen, Denmark. [5] Department of Clinical Biochemistry, Rigshospitalet, University of Copenhagen, Blegdamsvej 9, 2100 Copenhagen, Denmark. [6] NNF Center for Protein Research, Faculty of Health and Medical Sciences, University of Copenhagen, Blegdamsvej 3B, 2200 Copenhagen, Denmark. [7] Department of Pathology, Copenhagen University Hospital Hvidovre, Kettegaard Allé 30, 2650 Hvidovre, Denmark. [8] Gastrounit, Surgical Division, Copenhagen University Hospital Hvidovre, Kettegaard Allé 30, 2650 Hvidovre, Denmark. [9] Department of Nuclear Medicine and Functional Imaging, Ultrasound Section, Copenhagen University Hospital Hvidovre, Kettegaard Allé 30, 2650 Hvidovre, Denmark. [10] Department of Endocrinology, Copenhagen University Hospital Hvidovre, Kettegaard Allé 30, 2650 Hvidovre, Denmark. [11] Clinical Research Centre, Medical University of Bialystok, Jana Kilińskiego 1, 15-089 Białystok, Poland. [12] Department of Clinical Medicine, Faculty of Health and Medical Sciences, University of Copenhagen, Blegdamsvej 3B, 2200 Copenhagen, Denmark. [13] Department of Geriatrics, Copenhagen University Hospital Bispebjerg-Frederiksberg, Bispebjerg Bakke 23, 2400 Copenhagen, Denmark. [14] These authors jointly supervised this work: Steen Larsen, Flemming Bendtsen, Flemming Dela. ✉email: julie.steen.pedersen@regionh.dk

Non-alcoholic fatty liver disease (NAFLD), which is strongly linked to obesity and metabolic dysregulation such as type 2 diabetes (T2DM), has emerged as one of the most frequent causes of chronic liver disease across most of the world[1,2]. In NAFLD, fat accumulates within the hepatocytes but the disease spectrum of the condition is broad and gives rise to a variety of phenotypes, from the relatively benign non-alcoholic fatty liver (NAFL), to the aggressive non-alcoholic steatohepatitis (NASH), which may eventually progress to cirrhosis[3]. Despite extensive research, the pathophysiology driving the different phenotypes of NAFLD remains poorly understood[4,5]. Yet, hepatic mitochondrial dysfunction has been hypothesised to be involved in the progression of NAFL to NASH and fibrosis[6–9] and reduced mitochondrial oxidative phosphorylation (OXPHOS) capacity has previously been reported in liver tissue sampled from patients with obesity and NASH when compared with OXPHOS in those with NAFL[10].

Visceral- (VAT) and subcutaneous adipose tissue (SAT) -inflammation, -adipokine dysregulation and -insulin resistance are important defining features of obesity and major contributors to the development of obesity-related metabolic complications including NAFLD[11–16]. For instance, release of free fatty acids from insulin resistant adipose tissue is the primary source of the lipids accumulating in the liver in NAFLD[17,18] and adipose tissue inflammation has previously been associated with NAFLD severity[15,16].

Studies in humans and rodents have previously indicated suppressed mitochondrial biogenesis[19], downregulation of genes that encode mitochondrial respiratory complex products[20] and decreased respiration/OXPHOS[21] in adipose tissue in obesity.

Suboptimal mitochondrial function may be connected directly to both the adipose tissue dysregulation[22] and the development of metabolic disease[23]. Moreover, data from mice even suggest a link between impaired adipocyte mitochondrial function and decreased adiponectin (a recognised hepatoprotective and anti-fibrotic adipokine[24–26]) levels, decreased adiponectin synthesis[27], as well as hepatic steatosis[28].

If adipose tissue OXPHOS reflects adipose tissue metabolic dysregulation, the level of mitochondrial respiration could in turn influence NAFLD severity. Moreover, SAT and VAT do not present with the same mitochondrial respiratory capacity, as VAT has a lesser respiratory reserve capacity compared with SAT[29]. Thus, a decreased capacity to oxidise substrates in VAT may indirectly negatively influence the liver, because of potential lipid peroxidation, and cell damage and subsequent release of inflammatory cytokines/adipokines to the liver via the portal vein. In addition, a diminished capacity to oxidise lipids may result in increased release of FFA into the portal vein, thereby facilitating hepatic steatosis. Yet, no studies have measured VAT *and* SAT mitochondrial respiratory capacity together with hepatocyte mitochondrial respiratory capacity in patients with NAFLD. Also, the effect of major weight loss induced by bariatric surgery on hepatic OXPHOS is unexplored.

We investigated OXPHOS patterns using high-resolution respirometry (HRR) analyses in VAT, SAT, and liver tissue sampled from subjects with obesity and with varying stages of NAFLD who underwent bariatric surgery and in control study subjects without obesity (CON).

We predicted that mitochondrial respiration in VAT and SAT was mirrored in liver tissue and reflected NAFLD severity, with the lowest OXPHOS in all three tissues be measured in those with NASH.

In addition, we investigated the effects of bariatric surgery on hepatic OXPHOS capacities 12 months after surgery.

Here, we report no impact of NAFLD severity on hepatic OXPHOS in individuals with obesity, an obesity driven decrease in adipose tissue respiration but a marked increase in hepatic OXPHOS and hepatic mitochondrial biogenesis 12 months after bariatric surgery.

## Results

**Subjects characteristics at baseline (Table 1).** Weight, BMI and waist-hip ratio were similar across the NAFLD (NAFL−, NAFL+ and NASH) groups. Subjects in the NASH group were more likely to have T2DM ($P < 0.05$) and median (interquartile range, IQR) HOMA-IR was higher in NASH (7.8 (6.2─10.0) vs. NAFL− 4.0 (2.9─5.4), $P < 0.001$) and NAFL+ (5.3 (4.4─6.4), $P = 0.057$). Alanine aminotransferase (ALT) levels were higher in the NASH group (32 U/L (30─52) vs. NAFL− group (26 U/L (20─31)), $P < 0.05$) and NAFL+ group (27 (22─39), $P < 0.05$).

Mitochondrial content in the hepatic biopsies and in the adipose tissue was not different across CONs and NAFLD groups (Table 2). Figure 1 depicts the schematic study set-up.

**Oxygen fluxes in liver tissue (Figs. 2a–f, 4a+b and Supplementary Data S1).** In general, both mass-specific (Fig. 2a), citrate synthase (CS) activity (Fig. 2c) and mtDNA/nDNA -corrected oxygen fluxes (Fig. 2e) in liver tissue in substrate-inhibitor protocol 1 (SUIT P1) followed a specific pattern, with NAFL+ presenting with the highest numerically respiratory rates across all substrate steps, NASH and NAFL− with intermediate flux rates, and CON with the lowest oxygen flux rates. However, this was non-significant between the four groups.

First, we corrected mass-specific hepatic fluxes for variation in CS levels (Table 2), which did not alter the respiratory- or significance patterns in SUIT P1 (Fig. 2c). Respiratory rates remained non-significant between groups (Fig. 2c).

When adjusting for mtDNA/nDNA (Fig. 2e, Table 2 and Supplementary Data S1) OXPHOS$_{max}$ pr. mtDNA/nDNA (GMOS$_D$), the step representing maximal coupled oxidative phosphorylation, was significantly increased by 55% in NAFL+ when compared to CON ($P < 0.05$) and by 23% and 18% (albeit not significantly) compared with NAFL− and NASH, respectively. No significant differences between groups were observed for the maximal-uncoupled respiration (FCCP) ($P = 0.054$), GM$_D$ capacity ($P = 0.047$ but significance was lost in the pairwise comparison with correction for multiple comparison) or GMO$_D$ capacity ($P = 0.070$) between the NAFLD groups.

The P/E (Fig. 4a) and RCR (Fig. 4b) were also similar in the four groups.

In regression analyses, none of the chosen eight predictors (BMI, HOMA-IR, steatosis grade, ballooning grade, VAT OXPHOS$_{max}$, plasma leptin, plasma adiponectin and alanine aminotransferase) were found to be significantly associated with liver OXPHOS$_{max}$ per CS.

BMI was the only significant predictor (out of the above eight mentioned) of liver OXPHOS$_{max}$ per mtDNA/nDNA (unstandardised $\beta$: 0.0013; 95% CI: 0.0004─0.0021, adjusted $R^2 = 0.128$, $P < 0.01$). In the multiple linear regression of hepatic OXPHOS$_{max}$, using the four independent variables from the simple linear regressions with the lowest $P$-values (BMI, leptin ($P = 0.050$), ALT ($P = 0.081$) and steatosis grade ($P < 0.05$)), all were positively associated with hepatic OXPHOS$_{max}$,) but only BMI proved to be a significant predictor (unstandardised $\beta$: 0.0013; 95% CI: 0.0004─0.0021, adjusted $R^2 = 0.127$, $P = 0.004$). In post hoc multiple linear regression analyses neither sex ($P = 0.634$), T2DM status ($P = 0.258$) status or age ($P = 0.214$) were significantly associated liver OXPHOS$_{max}$ per mtDNA/nDNA and BMI remained statistically significant ($P = 0.002$) (Supplementary Information).

**Table 1 Clinical, anthropometrical and biochemical characteristics at baseline in patients with obesity (stratified by NAFLD severity) and normal weight control subjects undergoing bariatric surgery and cholecystectomy, respectively.**

| | Baseline | | | |
| | Patients with obesity according to histology | | | Controls |
| | NAFL− (n = 16) | NAFL+ (n = 15) | NASH (n = 9) | CON |
|---|---|---|---|---|
| Liver histology | | | | |
| NAFLD activity score | 2 (2-3) | 3 (3-4) | 5 (5-5)[‡‡‡,§§] | 1 (0.5-1.5)[§§,†††] |
| Steatosis | 0 (0-0) | 1 (1-1)[‡‡‡] | 2 (1-2)[‡‡‡] | 0 (0-0)[§§§,†††] |
| Inflammation | 1 (1-2) | 1 (1-1) | 2 (1-2)[‡] | 1 (0.5-1)[#] |
| Ballooning | 1 (1-1) | 1 (1-1.75) | 2 (2-2)[‡‡,§] | 0 (0-0.5)[‡‡,§§,†††] |
| Age (years) | 44 (36-51) | 45 (40-52) | 44 (41-53) | 39 (34-43) |
| Female (%) | 25 (81) | 8 (50) | 8 (53) | 7 (78) |
| RYGB (%)/SG | 12 (39)/19 | 12 (75)/4 | 5 (33)/10 | 0 |
| Diabetes (%) | 2 (5) | 5 (31) | 8 (53)[a] | 0 |
| Hypertension (%) | 8 (26) | 6 (37) | 5 (33) | 0 |
| Dyslipidemia (%) | 5 (16) | 6 (38) | 6 (40) | 0 |
| Weight (kg) | 121 (105-134) | 136 (118-168) | 127 (110-132) | 70 (62-80)[‡‡‡,§§§,†††] |
| BMI (kg/m$^2$) | 41.6 (37.1-45.2) | 45.6 (39.1-52.9) | 41.6 (35.4-47.1) | 24.5 (22.3-26.4)[‡‡‡,§§§,†††] |
| Waist-hip ratio | 0.83 (0.80-0.94) | 0.90 (0.87-0.95) | 0.97 (0.90-1.04)[‡‡‡] | 0.84 (0.75-0.90)[††] |
| ALT (U/L) | 26 (20-31) | 27 (22-39) | 32 (30-52)[‡,§] | 19 (18-27)[††] |
| AST (U/L) | 23 (20-28) | 21 (18-30) | 27 (20-30) | 20 (17-26) |
| Fasting glucose (mmol/L) | 5.8 (5.5-6.1) | 6.4 (5.5-8.4) | 6.5 (6.2-7.0)[‡] | 5.3 (5.2-6.0)[§,††] |
| C-peptide (pmol/L) | 1050 (922-1270) | 1280 (1060-1410) | 1655 (1270-2112)[‡‡] | 863 (650-921)[§§,†††] |
| Fasting insulin (pmol/L) | 104 (84-143) | 116 (103-172) | 191 (147-231)[‡‡,§] | 64 (47-96)[†††] |
| HbA1c (mmol/mol) | 35.0 (33.0-38.0) | 37.0 (35.0-50.0) | 37.5 (35.5-40.0) | 33 (29-34.5)[§,††] |
| HOMA-IR | 4.0 (2.9-5.4) | 5.3 (4.4-6.4) | 7.8 (6.2-10.0)[‡‡‡,##] | 2.1 (1.6-3.5)[§,†††] |
| HsCRP (mg/L) | 4.4 (1.6-7.4) | 6.8 (4.2-12.0) | 3.6 (2.3-7.8) | 0.9 (0.7-2.9)[‡,§§§,†] |
| Adiponectin (ng/mL) | 6463 (4830-8781) | 4936 (4487-6056) | 5255 (2943-6307) | 9327 (4021-12630) |
| Leptin (ng/mL) | 50 (27-99) | 50 (31-80) | 26 (18-60) | 13 (8-21)[‡‡‡,§§§] |
| sCD163 (ng/mL) | 687 (571-758) | 651 (592-705) | 759 (664-915) | 556 (492-618)[‡,†††] |
| sCD206 (ng/mL) | 325 (241-424) | 394 (329-497) | 378 (315-440) | 256 (214-510) |
| IL-1β (ng/mL) | 0.05 (0.02-0.07) | 0.07 (0.03-0.10) | 0.02 (0.01-0.04)[§§] | 0.02 (0.02-0.05) |
| IL-6 (ng/mL) | 0.94 (0.60-1.38) | 1.07 (0.86-1.50) | 1.00 (0.91-1.45) | 0.62 (0.25-1.18) |
| TNF-α (ng/mL) | 1.97 (1.59-2.34) | 2.12 (1.77-2.50) | 1.97 (1.59-2.10) | 1.71 (1.16-2.43) |

Data are presented as medians (IQR). P-values (2-sided) are Kruskal–Wallis with correction for multiple comparison or Chi Square test.
NAFLD non-alcoholic fatty liver disease, RYGB Roux-en-Y gastric bypass, SG sleeve gastrectomy, BMI body mass index, ALT alanine aminotransferase, AST aspartate aminotransferase, HbA1c glycated haemoglobin, HOMA-IR Homoeostatic Model Assessment for Insulin Resistance, HsCRP high-sensitivity C-reactive protein, sCD163 soluble cluster of differentiation 163, sCD206 soluble cluster of differentiation 206, IL-1β interleukin 1 beta, IL-6 interleukin 6, TNF-α tumour necrosis factor alpha.
‡, ‡‡, ‡‡‡ denotes statistical significance (P < 0.05, 0.01, 0.001, respectively) compared with NAFL−.
§, §§, §§§ denotes statistical significance (P < 0.05, 0.01, 0.001, respectively) compared with NAFL+.
†, ††, ††† denotes statistical significance (P < 0.05, 0.01, 0.001, respectively) compared with NASH.
#P = 0.052 compared with NASH.
##P = 0.057 compared with NAFL+.
aDenotes significantly more study subjects with type 2 diabetes in the NASH group.

**Table 2 Citrate synthase activity and mtDNA/nDNA count in liver-, visceral adipose- and subcutaneous adipose tissue.**

| | Baseline | | | |
| | NAFL− | NAFL+ | NASH | CON |
|---|---|---|---|---|
| Liver tissue | | | | |
| Citrate synthase activity (µmol g$^{-1}$ min$^{-1}$) | 10.5 (7.5-12.4) | 10.9 (7.8-13.8) | 11.0 (7.7-12.6) | 13.2 (9.9-14.5) |
| mtDNA/nDNA count | 470 (421-520) | 479 (436-519) | 508 (463-531) | 536 (417-709) |
| Visceral adipose tissue | | | | |
| mtDNA/nDNA count | 268 (248-308) | 267 (254-302) | 259 (218-287) | 292 (201-378) |
| Subcutaneous adipose tissue | | | | |
| mtDNA/nDNA count | 235 (197-276) | 263 (215-285) | 239 (198-281) | 254 (229-276) |

In substrate-inhibitor protocol 2 (SUIT P2) (Figs. 2b, d, f and Supplementary Data S1) respiratory fluxes (mass-specific, CS and mtDNA/nDNA adjusted) were similar across groups. However, with the addition of rotenone (inhibitor of complex I respiration) NAFL+ displayed slightly higher mtDNA/nDNA adjusted fluxes when compared with both NAFL− and NASH (P < 0.01 and P < 0.01, respectively) (Fig. 2f). With the addition of Antimycin A (blocking of electrons from complex III) the mtDNA adjusted fluxes were again markedly higher in NAFL+ when compared with NASH (P < 0.05).

**Visceral adipose tissue (Fig. 3a+c, Fig. 4a+b and Supplementary Data S1).** NAFL+ and NASH presented with reduced mass-specific oxygen fluxes across most steps (GM$_D$, GMO$_D$ and

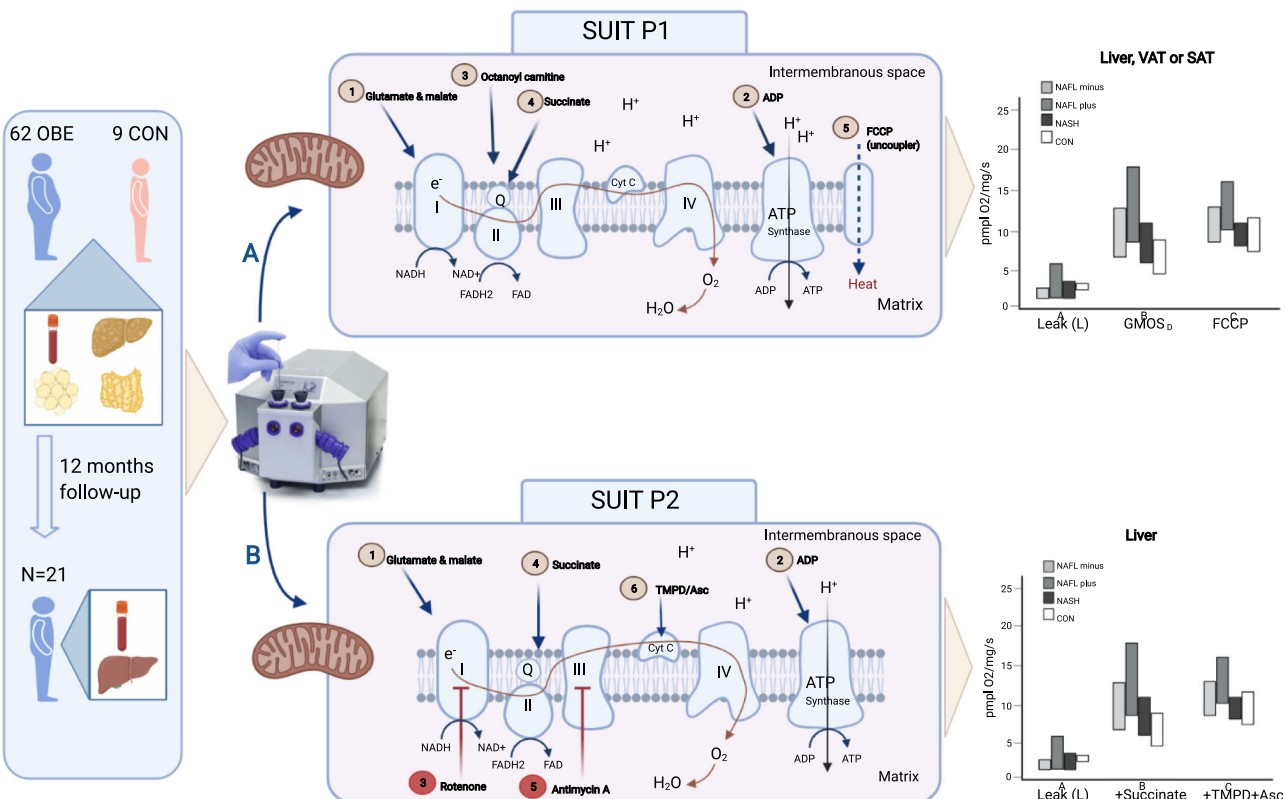

**Fig. 1 Depiction of the experimental study set-up.** Liver tissue, visceral adipose tissue (VAT) and subcutaneous adipose tissue (SAT) were sampled from 62 patients undergoing surgery—Roux-en-Y gastric bypass ($n = 29$) or sleeve gastrectomy ($n = 33$)—and stratified into three groups: 31 patients without liver steatosis (NAFL−); 16 patients with liver steatosis (NAFL+); and 15 patients with NASH (NASH). Nine normal weight control subjects without hepatic steatosis and who underwent planned laparoscopic cholecystectomy served as a control group (CON). Twenty-one patients with obesity underwent repeat liver biopsy 12 months after bariatric surgery. Fresh tissue from both timepoints underwent ex vivo, high-resolution respirometry (HRR) measurements in an Oxygraph, which measures tissue oxygen consumption and hence can determine mitochondrial respiratory chain complexes' (I–IV) respiratory rates and maximal electron transport system capacity. Two separate sequential substrate-inhibitor (SUIT) protocols were applied in the tissue. SUIT P1, panel **A**: (1) Adding of glutamate and malate (GM) to reveal leak respiration through complex I (2) Adding of ADP ($GM_D$) to initiate complex I linked oxidative respiratory process (3) Addition of a lipid (the fatty acid octanoyl carnitine) to assess complex II lipid respiratory capacity ($GMO_D$) (4) Addition of the tricarboxylic acid cycle substrate succinate to achieve maximal input of electrons through both complex I and complex II and maximal coupled oxidative phosphorylation ($GMOS_D$ or $OXPHOS_{max}$) (5) testing of maximal electron transfer system capacity (ETS) by stepwise addition of a protonophore (FCCP) to reveal uncoupled respiration (E). Before addition of FCCP cytochrome C was added as a quality control to test mitochondrial membrane integrity. Suit P2 (only liver tissue), panel **B** (1) glutamate and malate (GM, leak) (2) adding of ADP ($GM_D$) to initiate complex I linked oxidative respiratory process (3) Blocking of complex I by addition of rotenone followed by (4) addition of succinate to assess specific complex II respiration (5) blocking of electron transfer from complex III with antimycin A followed by (6) addition of the powerful stimulator N,N,N′,N′-tetramethyl-p-phenylenediamine (TMPD) and ascorbate to assess specific complex IV respiration. For further details of SUIT protocols, please see Methods and Supplementary Information. This figure was created with BioRender.com.

$OXPHOS_{max}$ and FCCP) when compared with CON (Fig. 3a). No differences among the NAFLD groups were found. Although mtDNA copy numbers were not statistically different between groups ($P = 0.624$) (Table 2), when correcting fluxes for mtDNA/nDNA content, respiratory fluxes were no longer significantly different (Fig. 3c).

sCD163, a plasma macrophage marker, was the factor most strongly and negatively associated with VAT $OXPHOS_{max}$ but it was neither significant in the univariable model ($P = 0.042$, $R^2 = 0.055$) nor in the multivariable model with BMI, sCD163, adiponectin and HOMA-IR ($P = 0.048$, $R^2 = 0.054$) with the significance level set to <0.006.

**Subcutaneous adipose tissue (Fig. 3b+d, Fig. 4a, b and Supplementary Data S1).** In SAT, mass-specific $OXPHOS_{max}$ was decreased by 39%, 35% and 35% in NASH, NAFL+ and NAFL−, respectively, when compared with CON ($P < 0.001$, $P < 0.001$ and

$P < 0.001$). Maximal-uncoupled respiration was comparably decreased by 39%, 37% and 36% ($P < 0.001$ for all NAFLD groups vs. CON) (Fig. 3b), in NASH, NAFL+ and NAFL−, respectively. After adjustment for mtDNA/nDNA the significance remained between CON and NAFLD groups (Fig. 3d).

High BMI did not predict low-SAT $OXPHOS_{max}$ per mtDNA/nDNA in univariable (unstandardised $\beta$: 0.0000136, 95% CI: −0.000110 to −0.000010, adjusted $R^2 = 0.097$, $P = 0.009$) or multivariable analyses with BMI, sCD163, adiponectin, or HbA1c (unstandardised $\beta$: 0.0000136 (−0.000116 to −0.000010), adjusted $R^2 = 0.093$, $P = 0.02$), according to the rigorous significance level of 0.006.

VAT and SAT P/E (Fig. 4a) were similar between groups but significantly increased ($P < 0.001$) compared with P/E in liver tissue, indicating that in AT the respiratory reserve capacity is less than in liver tissue.

RCR in VAT and SAT (Fig. 4b) were comparable among the four groups but markedly higher ($P < 0.001$) than in liver tissue.

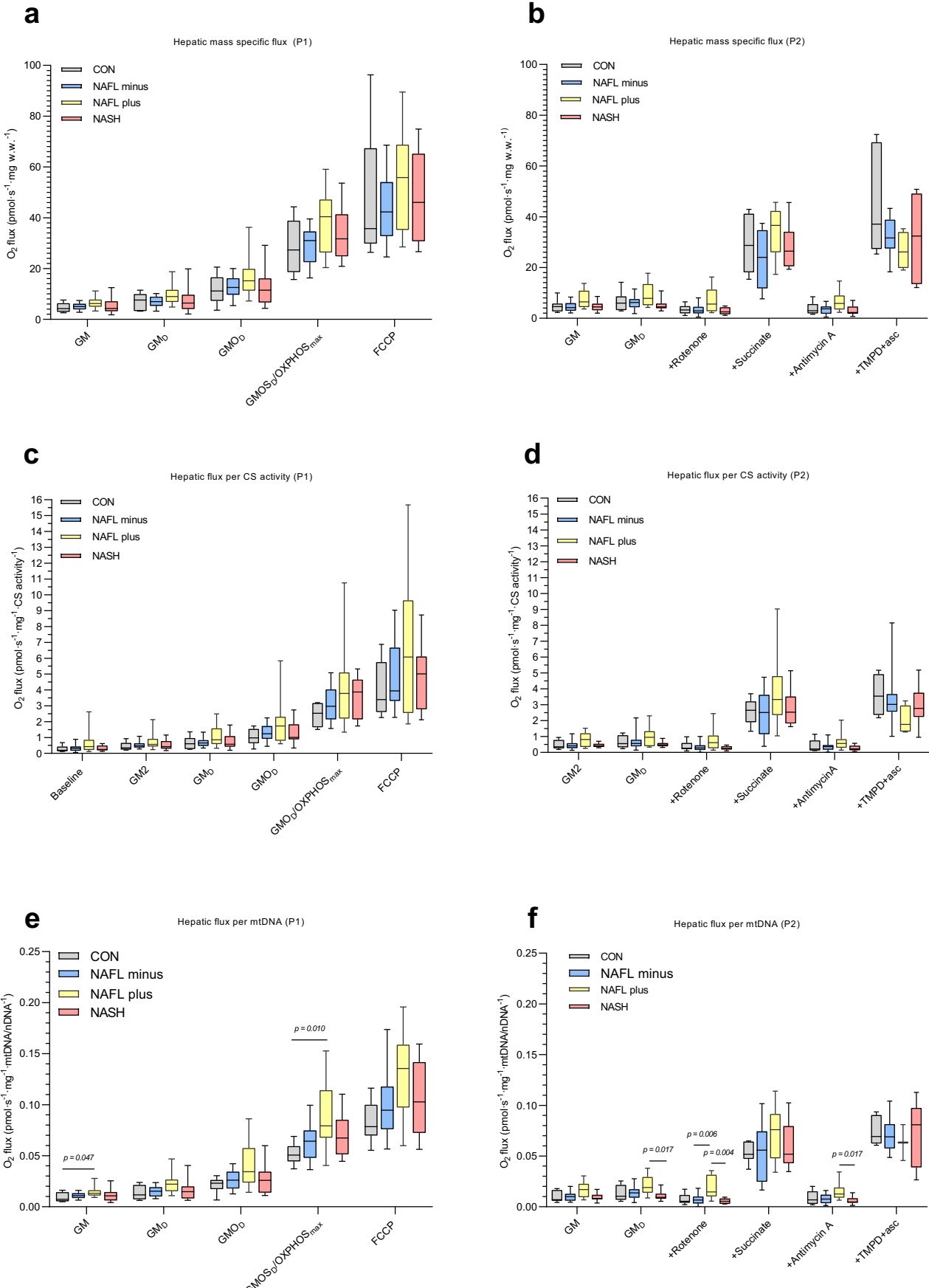

**Fig. 2 Mitochondrial respiratory rates in liver tissue at baseline. a** Mass-specific substrate-inhibitor protocol 1 (SUIT P1) $O_2$ fluxes. **c** P1 fluxes adjusted to citrate synthase activity (CS). **e** P1 fluxes adjusted to mitochondrial DNA/nuclear DNA (mtDNA/nDNA). **b** Mass-specific substrate-inhibitor protocol 2 (SUIT P2) fluxes $O_2$ fluxes. **d** P2 fluxes adjusted to CS. **f** P2 fluxes adjusted to mtDNA/nDNA. For details of SUIT protocols, please refer to Methods and Fig. 1. Data are median (horizontal line), interquartile range (boxes) and 10–90% percentile (error bars). P-values (two-sided) are Kruskal–Wallis pairwise comparison with correction for multiple comparison. Grey boxes; CON, blue boxes; NAFL−, yellow boxes; NAFL+, red boxes; NASH. SUIT P1: NAFL− $n = 30$, NAFL+ $n = 13$, NASH $n = 13$, CON $n = 6$. SUIT P2: NAFL− $n = 26$, NAFL+ $n = 16$, NASH $= 12$, CON, $n = 7$. Source data are provided as a Source Data file. Specific n's in each protocol and for each substrate and group are provided in the Source Data file.

Patients with and without T2DM had comparable respiratory fluxes in both VAT and SAT (Supplementary Data S3) and T2DM, age and sex were insignificant predictors of VAT/SAT OXPHOS in post hoc regression models (Supplementary Information). In logistic regression models neither OXPHOS$_{max}$ nor maximal-uncoupled respiration (FCCP) in VAT ($P = 0.449$, $P = 0.515$) or SAT (0.445, 0.713) predicted the presence of NASH in liver tissue in patients with obesity.

**Twelve months follow-up (Figs. 5 and 6, Table 3 and Supplementary Data S2).** Twelve months after surgery most clinical and biochemical parameters had improved. BMI dropped from 43.3 kg/m$^2$ (37.9–45.7) to 32.5 kg/m$^2$ (27.3–37.5) ($P < 0.001$). HOMA-IR (baseline: 4.4 (3.5–7.2), 12 months: 1.9 (1.4–2.4) ($P < 0.01$) decreased, and the plasma inflammatory profile was markedly improved (Table 3). NAS decreased by a median of two points (0.5–2), with significant reductions in steatosis, inflammation and ballooning (Table 3).

Hepatic maximal electron transport capacity had increased markedly 12 months after surgery ($P < 0.01$), as evidenced by the uncoupled respiratory capacity. The increase in OXPHOS$_{max}$ was not significant ($P = 0.099$) (Fig. 5a), but overall resulted in a significant reduction in the P/E ($P < 0.05$) (Fig. 5e), indicating an increased hepatic respiratory reserve capacity 12 months after bariatric surgery, which was not different from the capacity in the CON group.

Specific complex II and IV, but not complex I, activity was noticeably increased by 23% and 80%, respectively, after 12 months when compared to baseline ($P < 0.05$) (Fig. 5b). Median hepatic mtDNA/nDNA increased substantially by 32% (−12% to 51%) from a median count of 481 (441–527) to 660 (431–753) ($P = 0.012$) (Fig. 6a+b). When dividing the crude fluxes by mtDNA/nDNA to estimate 'respiration per mtDNA' the respiration was still significantly higher for pure complex IV ($P = 0.043$) (Fig. 5d).

Median P1/P2 mass-specific rates and P1/P2 fluxes per mtDNA/nDNA were not significantly different between SG vs RYGB operated individuals.

In a simple linear regression analysis total weight loss was not associated with the delta change in respiration per mtDNA/nDNA (lowest $P = 0.091$ for delta complex II activity. Unstandardised $\beta = -0.003$, 95% CI: −0.007 to 0.001, adjusted $R^2 = 0.205$). Type of surgery and delta NAS (improvement in NAFLD) were also not significant predictors of the increased respiration (Supplementary Information).

## Discussion

There are three major findings in this study. First, we found preserved hepatic respiratory rates in patients with NAFLD, including in those with NASH. Second, we found markedly decreased mass-specific SAT and VAT respiratory rates in the NAFLD groups. Third, we found significantly increased hepatic mitochondrial respiratory capacity as well as increased hepatic mtDNA/nDNA content 12 months after bariatric surgery, which

appeared to be driven primarily by a substantial increase in mitochondrial biogenesis.

For the measurements of hepatic tissue mitochondrial respiratory capacity, we employed two different 'SUIT' protocols (Figs. 1 and 2), generating independent measurements of mitochondrial respiration in two different specimens in duplicates from each patient. As such, we are confident in our conclusion that NASH does not result in changes in any of the assessed parameters (Leak (GM), GM$_D$, OXPHOS$_{max}$, ETS (FCCP), RCR or P/E) compared with CONs, which is in accordance with our earlier observations of patients with obesity with and without T2DM[30]. Though respiratory rates in those with NASH were numerically lower than those measured in NAFL +, no significance between these groups were found and NASH had persistently higher numerical fluxes than CON. Yet, compared with CON, patients in the NAFL + group did have significantly increased OXPHOS$_{max}$ ($P < 0.05$) but only when fluxes were corrected for mtDNA and not CS.

Our data suggest that mitochondrial respiratory function in human hepatic tissue is not affected by NASH and also not clearly augmented in those with NAFL+. This finding is overall not in line with conclusions reached in previous studies[10,31], but there is little consensus in the literature on this topic[6]. It has been proposed that hepatic energy metabolism including mitochondrial respiration increases with the development of steatosis[10,31], while decreasing markedly once NASH develops[10]. Koliaki et al.[10] drew this latter conclusion from their CS corrected flux data obtained in a cohort that was similar to the present, albeit including fewer patients in the NASH group.

Our conclusion that mitochondrial respiration *is not* decreased in liver tissue in patients with NASH stands in contradiction to the conclusion reached by Koliaki et al., who report significantly decreased intrinsic (respiration per CS) respiration in those with NASH compared with both NAFL+ and NAFL−[10]. Of note, respiration in those with NASH was not decreased when compared with their CON group. The authors speculate, however, that the decreased intrinsic respiration in NASH relative to NAFL+ and NAFL− is due to reduced 'metabolic adaptiveness' in the NASH liver tissue and reflective of mitochondrial dysfunction[10].

Some discrepancies appear between our study and the study by Koliaki and coworkers, the most important being that they recorded a 30–50% higher mitochondrial mass (measured by mode of CS per mg protein) in those with NASH compared with their CON, NAFL− and NAFL+ – groups. Thus, the diminished respiration in the NASH group in the study by Koliaki et al. is entirely driven by the reported increased mitochondrial content. In the present study, we did not at all observe a difference in mitochondrial mass when measured by mode of CS (Table 2). In particular, we found similar levels of CS within the NAFLD subgroups. To validate our findings of similar mitochondrial mass, we also quantified mtDNA/nDNA content and again found no significant differences in mitochondrial mass between groups (Table 2). Consequently, neither adjustment for CS nor mtDNA/nDNA changed our results and conclusion (Fig. 2e, f).

The second major finding was the decreased respiratory rates in VAT and SAT in NAFLD groups compared with CON

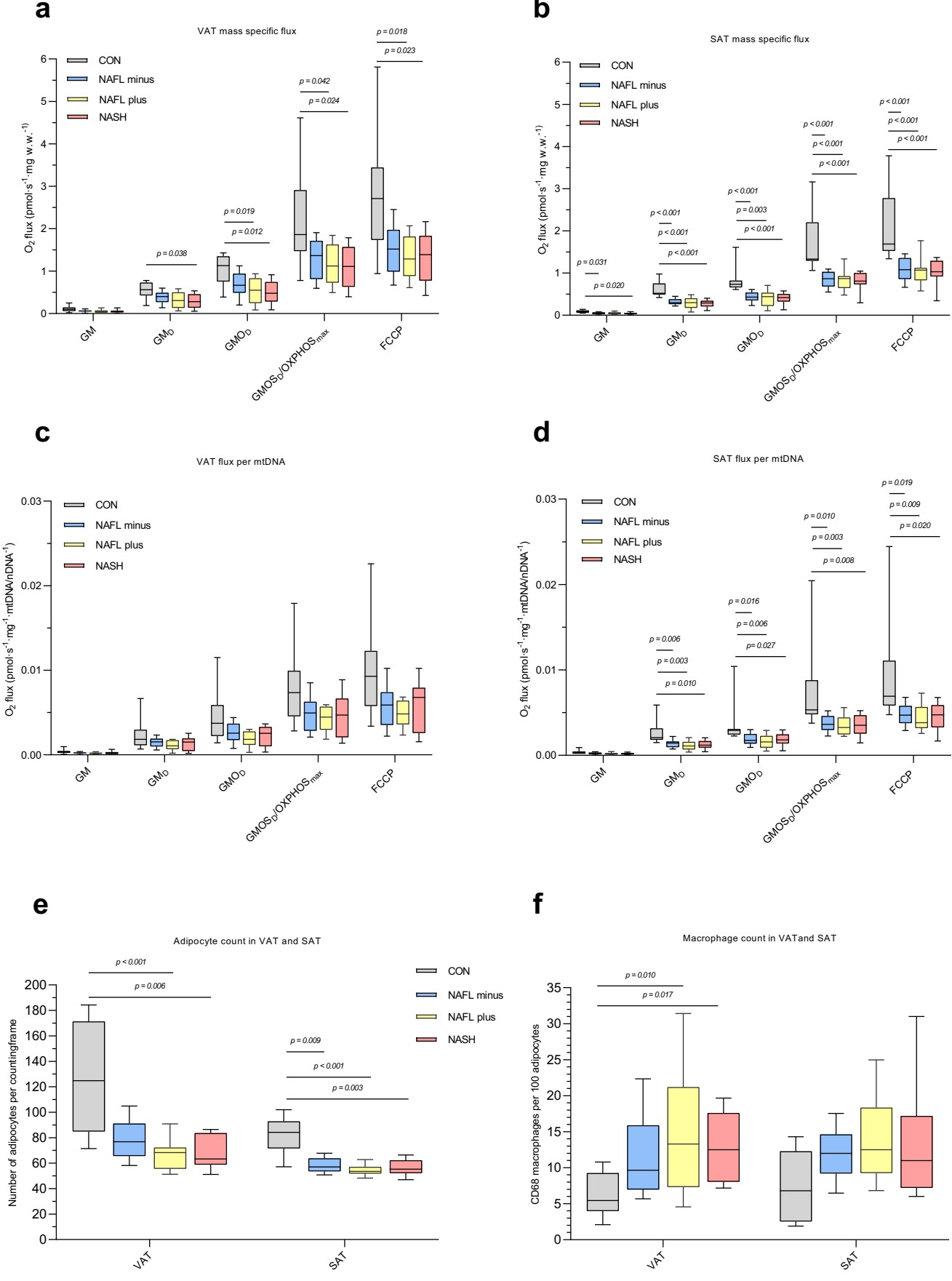

**Fig. 3 Mitochondrial respiratory rates in visceral- and subcutaneous adipose tissue (VAT and SAT, respectively) at baseline. a** Mass-specific substrate-inhibitor protocol 1 (SUIT P1) visceral adipose tissue (VAT) fluxes. **c** P1 VAT fluxes adjusted to mtDNA/nDNA. **b** Mass-specific substrate-inhibitor P1 subcutaneous adipose tissue (SAT) fluxes. **d** P1 SAT fluxes adjusted to mtDNA/nDNA. **e** Adipocyte count per counting frame (0.4 mm²) in VAT and SAT, respectively. **f** Number of CD68+ macrophage per 100 adipocytes in VAT and SAT. Data are median (horizontal line), interquartile range (boxes) and 10–90% percentile (error bars). *P*-values (2-sided) are Kruskal–Wallis pairwise comparison with correction for multiple comparison. Grey boxes; CON, blue boxes; NAFL−, yellow boxes; NAFL+, red boxes; NASH. VAT: NAFL− *n* = 30, NAFL+ *n* = 12, NASH *n* = 14, CON *n* = 9. SAT: NAFL− *n* = 29, NAFL+ *n* = 13, NASH *n* = 14, CON *n* = 9. Source data are provided as a Source Data file. Specific *n*'s for each substrate and group are provided in the Source Data file.

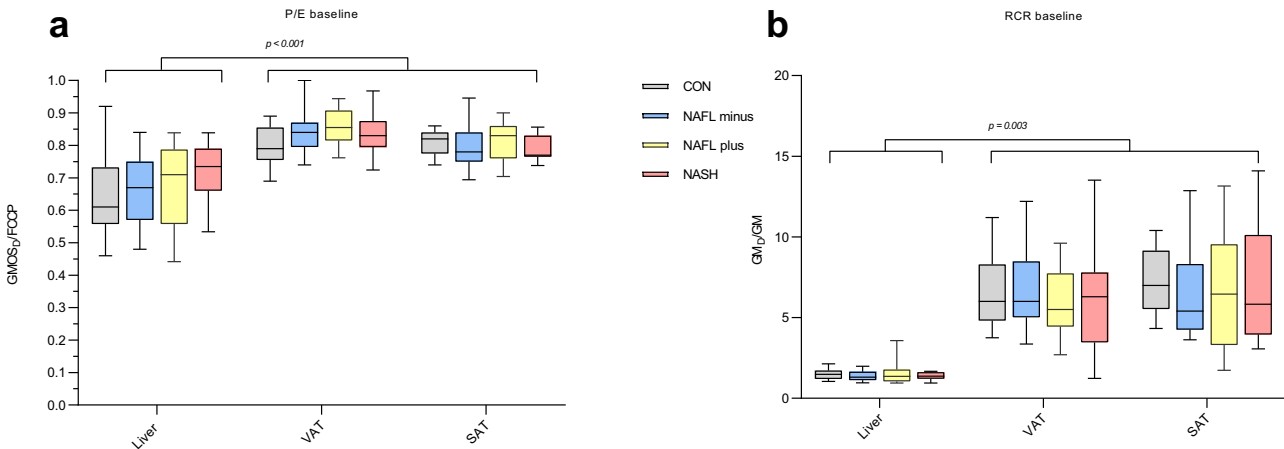

**Fig. 4 P/E and RCR in tissue. a** P/E (OXPHOS$_{max}$/FCCP) in liver tissue, visceral adipose tissue (VAT) and subcutaneous adipose tissue (SAT); **b** Respiratory control ratio (RCR), (GM$_D$/GM) in liver tissue, VAT and SAT. P/E and RCR were significantly higher (P/E: *P* < 0.001, RCR *P* = 0.003) in VAT and SAT when compared with liver tissue. No significant differences in P/E and RCR were found among the four groups in the three tissues. Data are median (horizontal line), interquartile range (boxes) and 10–90% percentile (error bars). *P*-values (2-sided) are Kruskal–Wallis pairwise comparison with correction for multiple comparison. Grey boxes; CON (*n* < 10), blue boxes; NAFL−, yellow boxes; NAFL+, red boxes; NASH. Source data are provided as a Source Data file. Specific *n*'s for each tissue and group are provided in the Source Data file.

(Fig. 3a–d), and for SAT this difference remained significant with adjustment to the mtDNA/nDNA ratio. This points towards a reduced mitochondrial capacity in SAT, that is not only explained by quantitatively-(fewer mitochondria) but also by qualitatively impairment in mitochondria function.

Our group has previously compared oxidative properties by mode of HRR in whole tissue VAT and SAT from humans with obesity and reported similar results as the present[29].

We are not aware of other AT mitochondrial respiration studies conducted among NAFLD patients that are available for comparison. In studies involving individuals with obesity data have primarily been obtained in SAT[32–34] and studies have assessed mitochondrial function by mode of mitochondrial gene expression profiles[20,33,34] with reports of impaired mitochondrial biogenesis and decreased expression of, e.g., genes encoding electron transport chain products. A few studies have compared VAT with SAT at a transcriptional level[20] and at a functional level in isolated mitochondria using enzymatic-based approaches[21,35]. In combination with data obtained in animal and experimental models of obesity, the evidence that obesity may be associated with a reduction in adipose tissue mitochondrial oxidative capacity and mitochondrial biogenesis is quite robust[19,22,36–38]. However, it is an ongoing debate whether the changes in mitochondrial function/reduced oxidative capacity are causative or secondary to the metabolic consequences (insulin resistance, T2DM) connected to obesity.

In relation to this debate, some interesting notions in our study deserve attention. Firstly, NAFLD severity did not appear to influence VAT or SAT intrinsic respiratory capacity (flux per mtDNA/nDNA) despite evidence of otherwise increasing

metabolic deterioration as witnessed for example by the high HOMA-IR in the NASH group. Secondly, we did not prove any correlation between AT OXPHOS and liver OXPHOS. Thirdly, nor did we find significant difference in OXPHOS when study subjects were stratified for presence of T2DM (Supplementary Data S3) nor when T2DM was a variable in the multiple linear regression analyses (Supplementary Information). Hence, the culprit of decreased AT respiration appears to be associated with obesity rather than with NAFLD or the systemic/tissue-specific metabolic deterioration such as insulin resistance, which NAFLD is reflective of.

This is in line with the findings of similar intrinsic mitochondrial respiratory capacity measured by mode of HRR in skeletal muscle[39] and hepatic tissue[30] from insulin resistant patients with T2DM compared with healthy controls. The findings are also in agreement with the conclusions reached by Chattopadhyay et al.[32] who found diminished respiratory chain activities (measured enzymatically) in isolated mitochondria from SAT sampled from individuals with obesity without any additional effect of T2DM.

Contrary to our initial predictions the largest discrepancies in mitochondrial respiration between CON and NAFLD subgroups were surprisingly found in SAT and not VAT, the latter of which is traditionally recognised as the fat depot with the strongest associations to metabolic disease including NAFLD[40–42]. Also, levels of CD68+ macrophages in both VAT and SAT in the NAFLD subgroups were indeed increased compared with CON, suggesting on-going adipose tissue inflammation, but macrophage counts were similar among both NAFLD groups and tissues (VAT vs. SAT) and degree of histological adipose tissue

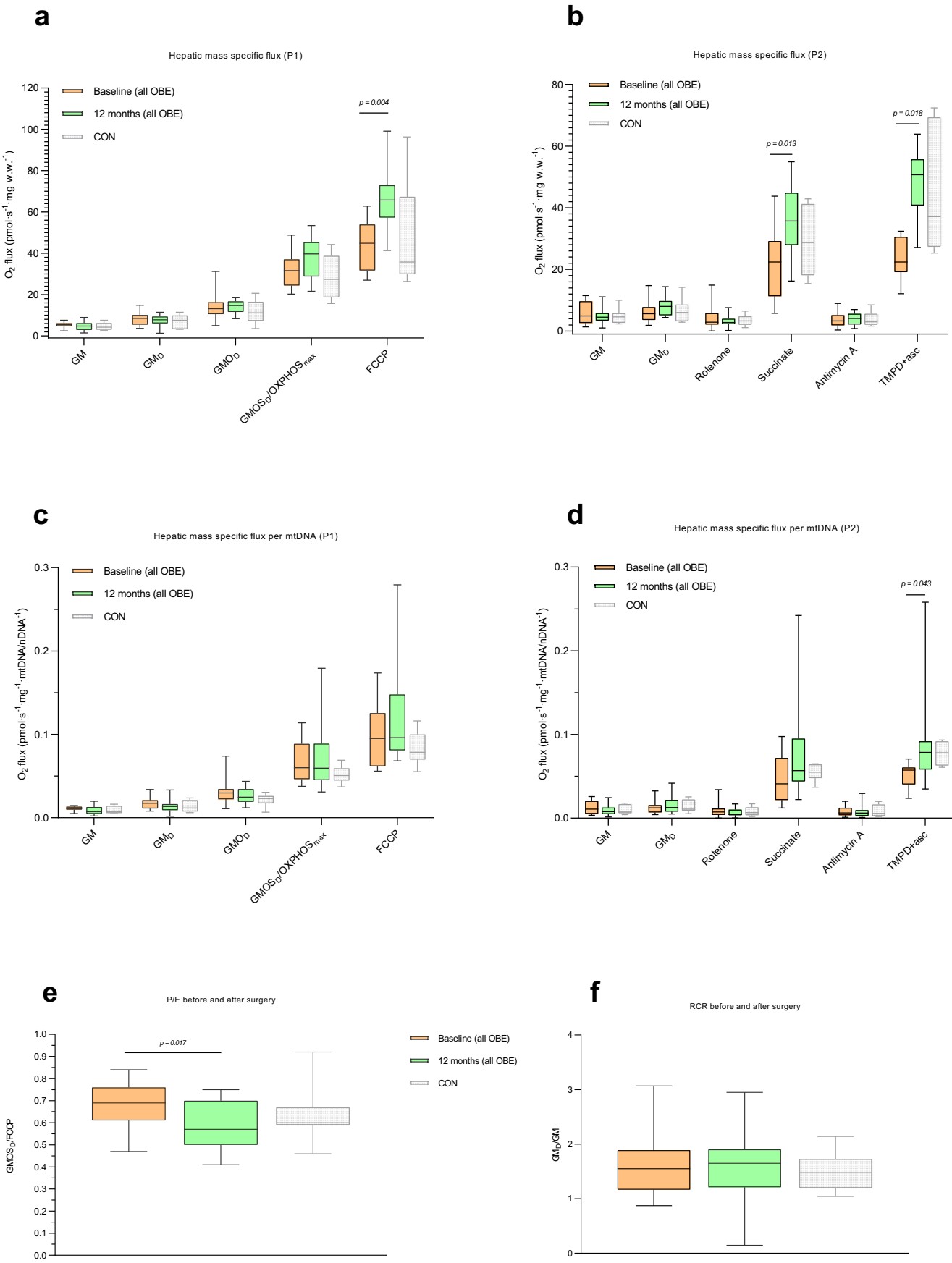

**Fig. 5 Mitochondrial respiratory rates in liver tissue obtained from 21 patients with obesity during Roux-en-Y gastric bypass surgery or sleeve gastrectomy (baseline) and again by Tru-Cut percutaneous liver biopsy twelve months later. a** Mass-specific substrate-inhibitor protocol 1 (P1) fluxes. **b** mass-specific protocol 2 (P2) fluxes; (GM$_D$/GM). **c** P1 fluxes adjusted to mtDNA/nDNA. **d** P2 fluxes adjusted to mtDNA/nDNA. **e** P/E (OXPHOS$_{max}$/FCCP). **f** Respiratory control ratio (RCR), (GM$_D$/GM) This patient cohort consisted of NAFL+ ($n = 6$), NAFL− ($n = 13$) and NASH ($n = 2$) (based on their baseline liver biopsy). CONs (baseline) are also shown. No statistically significant differences were found between CONs and OBE 12 months after surgery. P1 and P2 were performed as described in Fig. 1 and Methods. Data are median (horizontal line), interquartile range (boxes) and 10–90% percentile (error bars). P-values (2-sided) are Wilcoxon-signed rank test. Orange boxes; OBE baseline, green boxes; OBE 12 months after surgery, grey boxes (dimmed); CON ($n < 10$) at baseline. Source data are provided as a Source Data file. Specific $n$'s in each protocol and for each substrate, group and timepoint are provided in the Source Data file.

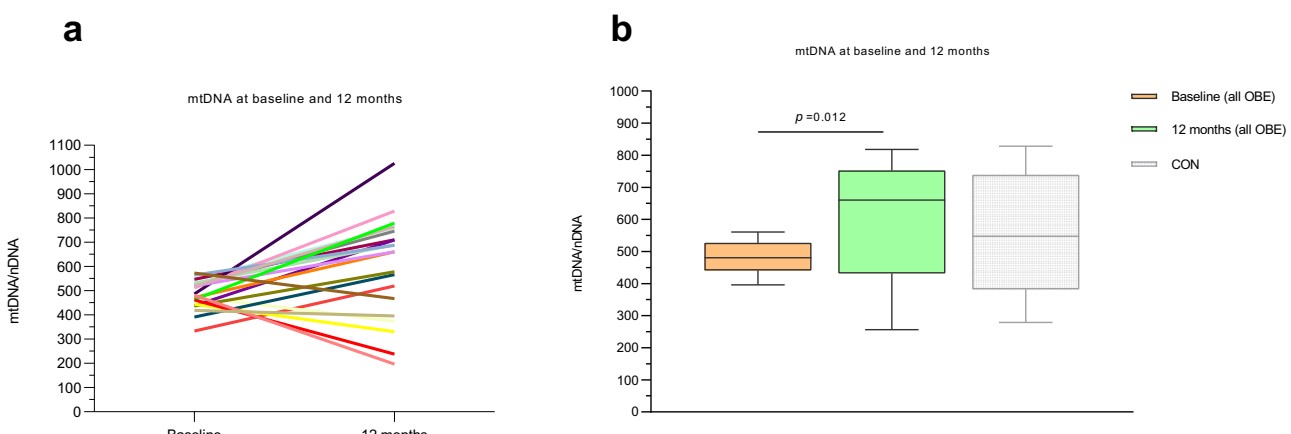

**Fig. 6 Mitochondrial DNA/nuclear DNA content in hepatic tissue from the 21 patients with obesity at baseline and 12 months later. a** spaghetti plot. **b** median mtDNA/nDNA at the two timepoints. CONs (baseline) are also shown. No statistically significant differences were found between CONs and OBE 12 months after surgery. Data are median (horizontal line), interquartile range (boxes) and 10–90% percentile (error bars). P-values (2-sided) are Wilcoxon-signed rank test. Orange boxes; OBE baseline, green boxes; OBE 12 months after surgery, grey boxes (dimmed); CON ($n < 10$) at baseline. Specific $n$'s in each protocol and for each substrate and timepoint are provided in the Source Data file.

inflammation was not significantly associated with low VAT or SAT OXPHOS$_{max}$ in regression models. Thus, and contrary to our initial hypotheses, we did not find a connection between decreased adipose tissue OXPHOS on one side and adipose tissue inflammation or NAFLD severity including presence of NASH on the other side.

It is a matter of speculation whether decreased adipose tissue OXPHOS is primarily due to increased adipocyte cell size, which is an undisputed observation in obese VAT and SAT sampled from individuals with obesity[40,43,44] and also reported in the present study (Fig. 3e). Consequently, fewer cells in AT per mg of wet tissue in the OBE vs. CON could lead to fewer mitochondria available for respiration. The size of adipocytes have been reported to both affect[45] and not affect[21] VAT and SAT mitochondrial OXPHOS capacities. However, adjustment of AT respiratory fluxes by mtDNA/nDNA content, which was similar between groups, approximates the respiration 'per cell' and in SAT respiration remained significantly lower in NAFLD groups. We therefore conclude that the decreased respiration is indeed intrinsically or quantitatively decreased in SAT.

The third major finding is that crude complex II respiration as well as both crude and mtDNA/nDNA-corrected complex IV respiration as well as ETS (FCCP), increased markedly 12 months after bariatric surgery (Fig. 5). Furthermore, P/E decreased significantly after 12 months (Fig. 5c), meaning that the hepatic mitochondria had gained extra reserve capacity during the 12 months after bariatric surgery. In fact, the P/E approached that seen in CON (Fig. 5e), even though OBE were still individuals with obesity compared with CON (BMI 32.5 kg/m$^2$ vs. 24.5 kg/m$^2$, respectively). The reported increase in hepatic respiration appears

to be driven by the marked increase in hepatic mtDNA/nDNA content of ~30%, suggesting that substantial and augmented mitochondrial biogenesis has taken place between baseline and 12 months after bariatric surgery.

We are not aware of any human studies for comparison. In mice/rat models using methodological approaches different to our own to assess mitochondrial function, RYGB and subsequent weight loss have been shown to increase mitochondrial respiratory chain capacity[46], CS activity[47], gene expression of respiratory chain complexes[46] and decrease oxidative stress[46,47] but not significant increase hepatic mtDNA[46].

The increase in hepatic respiration was irrespective of both mode of surgery, change in NAS and degree of weight loss, however, with only 21 study subjects we cannot exclude an effect of one or more of these parameters in a larger study cohort. Of note, the lack of association between weight loss and histological improvement on the one hand and increase in respiration on the other hand could also partly be explained by the uniformity of the data (the same degree of weight loss and same histological NAFLD improvement in all subjects) rather than the rather low number of study subjects 12 months after surgery. In our previously published paired data on liver histology in 40 individuals, 12 months after bariatric surgery we also could not conclude on the predictors for NAFLD improvement[48].

While the cross-sectional data comparison has many potential confounders (e.g., Table 1), the prospective paired comparison after 12 months has obvious strengths that makes the conclusion quite robust namely that following bariatric surgery there is an increase in hepatic mitochondrial respiratory capacity and mtDNA/nDNA content. As both mass-specific hepatic fluxes and

**Table 3 Clinical and biochemical characteristics at baseline vs. follow-up 12 months after bariatric surgery in 21 OBE study subjects.**

| | OBE baseline (n = 21) | OBE follow-up (n = 21) | P-value |
|---|---|---|---|
| NAFLD activity score | 3 (2–3.5) | 1 (0.5–2) | <0.001 |
| Steatosis grade | 0 (0–1) | 0 (0–0) | 0.08 |
| Inflammation grade | 1 (1–1) | 1 (0.5–1) | 0.035 |
| Ballooning grade | 1 (1–2) | 0 (0–0) | <0.001 |
| NAFL−/NAFL+/NASH | 13/6/2 | 18/3/0 | - |
| Age (years) | 44 (37–48) | NA | - |
| Sex (female/male) | 13/8 | NA | - |
| Type of surgery (SG/RYGB) | 13/8 | NA | - |
| Weight (kg) | 125 (116–137) | 90 (86–100) | <0.001 |
| BMI (kg/m²) | 43.3 (37.9–45.7) | 32.5 (27.3–37.5) | <0.001 |
| Waist-hip ratio | 0.86 (0.82–0.95) | 0.82 (0.78–0.88) | <0.001 |
| %EBWL | NA | 60 (47–84) | - |
| Total weight loss (kg) | NA | 32.7 (12.3–40.9) | - |
| Systolic blood pressure (mmHg) | 126 (119–138) | 112 (103–129) | 0.005 |
| Diastolic blood pressure (mmHg) | 84 (77–92) | 71 (65–82) | <0.001 |
| HR (bpm) | 71 (66–83) | 59 (50–70) | <0.001 |
| ALT (U/L) | 31 (25–39) | 23 (17–36) | 0.254 |
| AST (U/L) | 23 (18–34) | 24 (27–36 | 0.631 |
| Fasting glucose (mmol/L) | 5.9 (5.5–6.4) | 5.3 (5.0–5.5) | <0.001 |
| C-peptide (pmol/L) | 1160 (979–1360) | 779 (652–846) | <0.001 |
| Fasting insulin (pmol/L) | 128 (101–178) | 59 (41–72) | 0.002 |
| HbA1c (mmol/mol) | 35 (33–38) | 33 (32–36) | 0.002 |
| HOMA-IR | 4.4 (3.5–7.2) | 1.9 (1.4–2.4) | 0.002 |
| HsCRP (mg/L) | 4.4 (1.6–7.3) | 0.9 (0.4–2.0) | <0.001 |
| Adiponectin (ng/mL) | 5842 (4732–7695) | 11,979 (7827–13,689) | <0.001 |
| Leptin (ng/mL) | 45 (33–71) | 16 (7–33) | <0.001 |
| sCD163 (ng/mL) | 692 (603–749) | 589 (512–630) | 0.013 |
| sCD206 (ng/mL) | 336 (278–449) | 278 (227–433) | 0.018 |
| IL-1β (ng/mL) | 0.05 (0.03–0.08) | 0.04 (0.02–0.06) | 0.210 |
| IL-6 (ng/mL) | 0.96 (0.60–1.35) | 0.59 (0.34–0.84) | <0.001 |
| TNF-α (ng/mL) | 1.75 (1.47–2.28) | 1.93 (1.44–2.41) | 0.970 |

Data are presented as medians (IQR). P-values (2-sided) are Wilcoxon-signed rank test. NAFLD non-alcoholic fatty liver disease, SG sleeve gastrectomy, RYGB Roux-en-Y gastric bypass, BMI body mass index, %EBWL percentage excess body weight loss, mmHg millimetres of mercury, HR heart rate, bpm beats per minute, ALT alanine aminotransferase, AST aspartate aminotransferase, HbA1c glycated haemoglobin, HOMA-IR Homoeostatic Model Assessment for Insulin Resistance, HsCRP high-sensitivity C-reactive Protein, CD163 soluble cluster of differentiation 163, sCD206 soluble cluster of differentiation 206, IL-1β interleukin 1 beta, IL-6 interleukin 6, TNF-α tumour necrosis factor alpha.

fluxes/mtDNA increased it could be suggested that the augmented respiration is brought about via a combination of increased mitochondrial biogenesis/mitochondrial mass and increased intrinsic capacity of 'each' mitochondria. Altogether, our data are indicative of substantial metabolic flexibility of the human liver in response to bariatric surgery. The clinical importance of this finding adds to the existing knowledge on the extensive re-generative potential of the human liver as we here show that this capacity is present at the mitochondrial level, an essential basis for all metabolic processes.

This study benefits from a large data set with paired liver and adipose tissue samples from 71 study subjects and re-evaluation of hepatic mitochondrial oxidative capacity and mtDNA/nDNA content 12 months after bariatric surgery, which have not previously been carried out in humans. Another strength of the study is its analytical method, based on direct measurements of mitochondrial activity in paired tissue biopsies from the same study subject.

Indeed, ex vivo measurements of mitochondrial respiration can only be an estimate of the true, biological function. There are no methods available by which mitochondrial respiration can be measured in vivo, with the same degree of details, which can be obtained with the HRR. For skeletal muscle a crude comparison between oxygen consumption measured in biopsies with HRR can be made against measures of Phosphocreatine—Creatine system by use of nuclear magnetic resonance. The readouts are not the same, but Layec et al[49]. have shown the same directional changes in the readouts of the two methods.

For the liver, substrate metabolism can be measured in vivo by, e.g., 31 phosphorous nuclear magnetic resonance, and oxygen consumption by dynamic oxygen 17 ($^{17}$O) nuclear magnetic

resonance spectroscopy (NMR). However, the resolution is not near the resolution one can obtain with the ex vivo technique.

The main limitation of this study is its cross-sectional design at baseline, which does not allow for conclusions about the causality of the observed flux rates and their associations. Furthermore, the 8% mandatory weight loss required by the Danish Health Authorities for surgery could have impacted our results at baseline as our subjects were not weight stable. Yet, previously we have not found evidence of effect of the 8% diet induced weight loss on mitochondrial respiration in SAT[50].

There are also some limitations to the paired data as the first liver biopsy was obtained as a wedge biopsy shortly after the introduction of anaesthesia, while the percutaneous biopsy was obtained by needle under local anaesthesia (lidocaine) and penetrates deeper into the tissue. The influence of general anaesthesia on mitochondrial respiration is unknown. However, in a supplementary experiment in a pig liver (Supplementary Fig. S1), we could not demonstrate marked differences in respiration in relation to the site of biopsy or by mode of sampling (wedge vs. percutaneous). If anything, the respiratory rates were slightly lower in the percutaneous biopsy compared with the wedge biopsy, which therefore only strengthens the finding of increased respiratory rates after weight loss. Another limitation is the lack of re-sampling of adipose tissue at follow-up. Finally, it should be noted that mature adipocytes constitute between 31% and 57% of the cell types in human adipose tissue samples as recently shown by the use of spatially resolved transcriptional profiling with single-cell RNA sequencing[51].Therefore, the respiratory data reported here pertain to adipose tissue, and not necessarily specifically to adipocytes.

In conclusion, we report no evidence of impaired hepatic mitochondrial respiration in patients with obesity and NASH. However, in VAT and in SAT, especially, intrinsic respiration was lowered but the reduction was irrespective of NAFLD status; rather, obesity appeared to be the common denominator affecting respiration in a similar manner in both VAT and SAT. Thus, compromised adipose tissue respiration is not likely to significantly impact NAFLD severity. Whereas OXPHOS in liver tissue was generally increased in the individuals with obesity, OXPHOS was simultaneously decreased in AT, which point towards a differentiated respiratory adaptiveness in liver and AT in obesity but with little additive effect of increased metabolic strain. Moreover, our results of enhanced hepatic OXPHOS and increased hepatic mitochondrial biogenesis 12 months after bariatric surgery expand upon previous data that suggest the human liver is an extremely adaptable organ both during and after major weight loss and remission of NAFLD.

## Methods

**Ethics**. The study protocol was approved by the Scientific Ethics Committee in Capital Region of Denmark (H-16030784 and H-16030782). The Danish Data Protection Agency approved the collection, handling and storage of data (P-2019-514 and P-2020-606). Informed oral and written consent was obtained from all study participants. The study was conducted according to the Declaration of Helsinki.

**Study participants**. The respirometry data presented in the current study are based on 71 subjects in total: 62 patients with severe obesity (OBE) (41 females, 21 males, median age 44 years (IQR 39–57), undergoing laparoscopic bariatric surgery (either Roux-en-Y gastric bypass (RYGB) (n = 29) or sleeve gastrectomy (SG) (n = 33)) and nine normal weight control subjects (CON) (7 females, 2 males, median age 39 years (IQR 24–43). Subjects were enrolled between December 2016 and September 2019 at Copenhagen University Hospital Hvidovre. OBE fulfilled the existing criteria for bariatric surgery issued by the Danish Health Authorities, including a mandatory weight loss of 8% before surgery. Study-specific exclusion criteria were current or previous alcohol consumption of >2.5 units/day for men and >1.5 units/day for women, preexisting liver disease other than NAFLD, preexisting disease in the lipid metabolism and acute or chronic inflammatory disease,

or an ethnic origin other than North European. Mode of surgery (RYGB or SG) was decided by the endocrinologists at the Endocrinology Department. Additional information of criteria for bariatric surgery in Denmark can be found in Supplementary Information.

CON consisted of patients with a BMI between 18 and 27 kg/m$^2$ undergoing laparoscopic cholecystectomy due to gallbladder stones. CONs had to be otherwise healthy with no current or former comorbidities and zero intake of prescribed medication other than medication for mild allergies, migraine and/or contraception.

CONs received a fee of 1500 DKK (corresponding to 200 euros) before taxes (taxes of ~37% were applied).

**Study design and anthropometrics.** Blood samples were collected in the morning on the day of surgery (baseline) and after 12 months on the day of the percutaneous liver biopsy, on both occasions after a minimum of 10 h of fasting (Fig. 1). Study participants were phenotypically and anthropometrically characterised (Table 1).

During surgery (bariatric or cholecystectomy) a wedged liver biopsy and a specimen of VAT was sampled from the greater omentum. SAT was sampled from the abdominal trocar incision hole (upper left side of the abdomen). All biopsies were obtained immediately after trocar placement and before the surgical procedure.

Twenty-one OBE subjects underwent ultrasonically guided, percutaneous liver biopsies 12 months after surgery. Three pieces of tissue were sampled per patient. A flowchart of study flow is available (Supplementary Fig. S2).

At baseline the tissues were immediately divided into smaller pieces of 50–100 mg, placed in separate tubes containing a chilled mitochondrial preservation buffer (BIOPS)[29], snap-frozen in liquid nitrogen and stored at −80 °C for later mitochondrial DNA (mtDNA) analysis or immersed and fixated in 2% paraformaldehyde for paraffin embedment and histological evaluation.

At 12 months, the percutaneous liver biopsies were placed in BIOPS (HRR measurements) and paraformaldehyde (histology) or snap frozen for biobank (mtDNA analysis).

Tissues for measurements of mitochondrial respiratory capacity were transported on wet-ice and reached the laboratory 45–60 min after sampling.

**Mitochondrial respirometry protocols.** HRR allows direct, ex vivo measurement of mass-specific mitochondrial respiratory capacity. In brief, HRR measures the use of oxygen during electron flow through the complexes of the mitochondrial electron transport chain in response to the addition of substrates and inhibitors.

Each tissue biopsy was carefully trimmed while in fresh, chilled BIOPS and washed in respiration medium. Between 2 and 4 mg of liver tissue and 40 mg of both VAT and SAT were transferred to the 37 °C chamber, (liver tissue: hyperoxygenated (450–200 nmol/mL, AT: normooxygenated (200–100 nmol/mL) in the respirometer (Oroboros Instruments, Innsbruck, Austria. Software Oroboros DatLab 6.0) as described previously;[29,30] digitonin was added for permeabilization of adipose tissue. All tissues were studied in duplicate at baseline and liver tissue in singlicates after 12 months. Oxygen consumption was corrected for tissue wet weight and given as oxygen flux as pmol mg wet weight$^{-1}$ s$^{-1}$.

Two separate substrate-inhibitor (SUIT) protocols were used.

AT and hepatic tissue were studied in substrate-inhibitor (SUIT) protocol 1 (P1). State 2 leak respiration (GM) was measured after the addition of 10 mM glutamate and 2 mM malate and electron flow through complex 1. With no adenosine diphosphate (ADP) yet added, this step measured the oxygen consumption used for compensating the naturally occurring electron leak through the inner mitochondrial membrane. State 3 complex I respiration (GM$_D$) was obtained after adding ADP + MgCl$_2$ (5 mM + 3 mM). When ADP is added the oxidative respiratory process is initiated and ATP is produced. State 3 complex I and II lipid respiratory capacity was measured after the further addition of 1.5 mM of the medium chain fatty acid octanoyl carnitine (GMO$_D$), which stimulated mitochondrial β-oxidation and electron transport through complex I and subsequently fed electrons to complex II via the electron shuttle ubiquinone.

Subsequent addition of 10 mM of the tricarboxylic acid cycle (TCA) substrate succinate (GMOS$_D$) yielded the maximal coupled oxidative respiratory capacity (OXPHOS$_{max}$) through both complex I and complex II. Testing of mitochondrial outer membrane integrity was carried out with supplementation of 10 μM cytochrome C (CytC). A CytC response with a less than 10% increase from the GMOS$_D$ step was deemed acceptable and used as an indicator for preserved outer mitochondrial membrane not damaged from the preparation procedures. Lastly, by titrating the uncoupling agent p-triflouromethoxyphenylhydrazone (FCCP step) in steps of 0.2 μM we assessed maximal-uncoupled respiration (also called maximal electron transport system capacity (ETS)).

Hepatic tissue was also studied using substrate-inhibitor SUIT protocol 2 (P2). Firstly, oxidative phosphorylation was stimulated as described in P1 by addition of malate (2 mM) and glutamate (10 mM) (GM), and then ADP + MgCl$_2$ (5 mM + 3 mM) (GM$_D$), to yield complex I oxygen flux. By subsequently adding rotenone (0.5 mM), an inhibitor of complex I, and then the complex II substrate succinate (10 mM), specific complex II oxygen flux was assessed. By blocking electron flux through and from complex III with antimycin A (5 μM) and subsequently stimulating complex IV (the cytochrome-c oxidase, COX) with the COX-specific electron donor N,N,N',N' -tetramethyl-p-phenylenediamine (TMPD) (0.5 mM) and

ascorbic acid (2 mM), specific complex IV activity could be measured. Figure 1 provides a schematic overview of the study set-up and SUIT P1 and P2.

After thorough assessment of the raw data files we removed eight liver (five baseline and three follow-up), three VAT and one SAT HRR measurements from the analyses due to uncertain quality (e.g., high cytochrome c response). This left us with 21 subjects in total at 12 months with good quality hepatic data at both baseline and 12 months.

**Histological evaluation of liver and adipose tissue.**
*Liver tissue.* Sections of formalin fixed, paraffin embedded liver biopsies were stained with Hematoxylin & Eosin and Picrosirius Red. NAFLD activity score (NAS) was calculated using standard guidelines[52]. All biopsies were evaluated by three expert liver pathologists with specific expertise in NAFLD and who were blinded to clinical details. Consensus was sought in the event of disagreement among scores.

*Adipose tissue.* VAT and SAT were fixed in formalin and embedded in paraffin. From the paraffin blocks, 3 μM sections were cut and mounted onto coated slides (FLEX IHC slides K8020, Dako, Glostrup, Denmark). The sections were dried at 60 °C for 20 min. The slides were then placed on the OMNIS Platform (Agilent), deparaffinized on-board and submitted to heat induced epitope retrieval (HIER) in Flex TRS high pH (EnVision FLEX Target Retrieval solution, GV804) for 30 min at 97 °C. Following the primary antibody for CD68 (Monoclonal mouse antibody, clone KP1, Ready-to-use, GA609, Dako, Glostrup, Denmark) was applied for 25 min at 32 °C and endogenous peroxidase blocking (EnV Perox GV800/GV823). After a wash in buffer the visualisation system, OptiView DAB (EnV HRP-labelled polymer, GV800/GV823) was applied. The slides were finally developed with DAB (EnV FLEX Substrate Working Solution, GV825) and counterstained with hematoxylin (GC808). Slides were scanned using NanoZoomer (Hamamatsu, Hamamatsu, Japan) and evaluated digitally with the NDP.view2 viewing software (Hamamatsu, Hamamatsu, Japan) on a 2 K screen at ×20 magnification. Adipocytes and macrophages were counted in eight randomly chosen whole areas (each representing an area of 418.660 μm$^2$) per slide using an unbiased counting frame. Counting was performed blindly by one observer. Macrophages are expressed as number of CD68 + cells normalised to 100 adipocytes.

Based on liver histology sampled at baseline (during surgery), we stratified the OBE into three groups:
1) NAFL minus (NAFL−): 31 patients without liver steatosis and with a median NAS of 2;
2) NAFL plus (NAFL+): 16 patients with liver steatosis and a median NAS of 3;
3) NASH: 15 patients with a NAS of 5 or above and all with points from all NAS subcategories (steatosis, inflammation and ballooning).
CON had zero hepatic steatosis and a median NAS of 1.

In total, 21 OBE with repeat liver biopsies 12 months after surgery were available for our paired experiment. All 21 had valid P1 HRR runs at both timepoints (baseline and 12 months). The 21 OBE consisted of 13 with NAFL− at baseline, six with NAFL+ and two with NASH at baseline.

Phenotypically, anthropometrically, biochemically and in terms of OXPHOS and median NAS the 21 OBE with paired HRR analyses did not separate statistically significant from the 41 OBE who only participated in the cross-sectional analysis at baseline.

**Plasma biomarkers.** Routine metabolic and liver markers were analysed immediately after blood sampling. Plasma alanine aminotransferase (ALT), aspartate aminotransferase (AST), plasma-glucose and triglycerides were measured using the Roche/Hitachi Cobas c 8000 system (Roche Diagnostics GmbH, Mannheim, Germany) with Cobas calibrators and reagents, according to the manufacturer's instructions. Serum insulin and c-peptide concentrations were measured by immunoassay Cobas e 602. HbA1c was measured in plasma with the Tosoh TSKgel G8 Variant His on the Tosoh Automated Glycohemoglobin Analyzer HLC-723G8 (Tosoh Corporation, Tokyo, Japan). All markers were measured in the fasting state.

*Plasma inflammatory markers.* Frozen plasma was stored in the −80 °C freezer until analysis of a panel of selected inflammatory markers (Table 1). Adiponectin (Cat. No. DRP300, R&D Systems, McKinley Place NE, Minneapolis), sCD163 (Cat. No. DC1630, R&D Systems, McKinley Place NE, Minneapolis) and sCD206 (Cat. No. DLP00, R&D Systems, McKinley Place NE, Minneapolis) were measured by sandwich ELISA (Cat. No. ELH-MMR, RayBiotech, Norcross, Georgia). IL-1β, IL-6 and TNF-α were measured by customised electrochemiluminescence (ECL) assay (Cat. No. K151A9H-1, Mesoscale (MSD), Rockville, Maryland). sCD163 was measured in Hep-Plasma and the remaining inflammatory markers in cooled EDTA plasma. All markers were measured in the fasting state.

**Quantification of mitochondrial activity in VAT, SAT and liver tissue.** Citrate synthase (CS) activity: In liver tissue we measured CS activity and adjusted respiratory rates accordingly. CS activity was measured using spectrophotometry. Approximately 15 mg wet weight of liver tissue was homogenised in 600 μL 0.3 M K$_2$ HPO$_4$, 0.05 % BSA, pH 7.7 for 2 min on a Tissuelyzer (Qiagen, Venlo, Limburg,

Netherlands). 6 µL of 10 % Triton was added and the samples were left on ice for 15 min before they were stored at −80 °C for later analysis.

The homogenate was diluted 50 times in a solution containing 0.33 mM acetyl-CoA, 0.157 mM DTNB, 39 mM Tris-HCl (pH 8.0), final concentrations. The change in 5,5′-Dithiobis-(2-nitrobenzoic acid (DTNB) to 2-nitro-5-thiobenzoate anion (TNB) at 37 °C was measured spectrophotometrically at 415 nm after addition of 0,6 mM oxaloacetate on an automatic analyzer (Cobas 6000, C 501, Roche Diagnostics, Mannheim, Germany). Enzyme activities are expressed as micromoles substrate per minute per gram wet weight of liver tissue.

**Quantification of mitochondrial DNA**. In VAT, SAT and liver tissue we quantified mitochondrial DNA (mtDNA) content, which is presented as mtDNA/nuclear DNA (nDNA) ratio. Respiratory rates were adjusted accordingly.

To isolate DNA, 5–20 mg of adipose or liver tissue was disrupted in 250 µL of alkaline lysis solution (25 mM NaOH and 2 mM EDTA) using a QIAGEN TissueLyser II bead homogeniser with two consecutive one-minute rounds at 30 hertz. After homogenisation, samples were heated for one hour at 96 °C, and then neutralised with 250 µL of neutralisation buffer (40 mM Tris-HCl). After centrifugation at 12,000 × $g$, the supernatant was transferred to a new tube and used for semi-quantitative polymerase chain reaction (PCR). mtDNA/nDNA data were analysed as described[53], with the addition of primer efficiency correction. Two sets of primers were used to amplify mtDNA at the mitochondrially encoded cytochrome C oxidase II (mtCO2) and mitochondrially encoded cytochrome B (mtCYTB) loci, while primer sets in the peroxisome proliferator-activated receptor gamma (PPARG) and uncoupling protein gene 1 (UCP1) gene loci were used to amplify nuclear DNA. Primer sequences: mtCO2 F: tgaagccccattcgtataa, mtCO2 R: cgggaattgcatctgtttttt, mtCYTB F: agacagtcccaccctcacac, mtCYTB R: ggtgattcctaggggg gttgt, nUCP1 F: gcccaatgaatactgccact, nUCP1 R: tgcatgcattctaggtctttaattt, nUCP1 R: tgcatgcattctaggtctttaattt, nPPARG F: ttcagaaatgccttgcagtg, nPPARG R: acattttttggcaat ggcttt.

Primer efficiency was determined by running serial dilution curves and analysing the slope with Lightcycler 480 Software (Roche Life Science). Efficiencies were determined to be 1.938 (93.8%) for mtCO2, 1.923 (92.3%) for mtCYTB, 1.839 (83.9%) for nUCP1 and 1.892 (89.2%) for nPPARG.

Efficiency corrected Ct values (Cteff) were calculated by the equation $\log_2((1 + Eff)^{Ct})$, where Eff is the efficiency for the gene (for example, Eff=1.938 for mtCO2). The delta Ct method was used to calculate DNA levels for each mitochondrial gene relative to each nuclear gene using the equations $2^{(Cteff\ nPPARG\ -\ Cteff\ mtCO2)}$, $2^{(Cteff\ nUCP1\ -\ Cteff\ mtCO2)}$, $2^{(Cteff\ nPPARG\ -\ Cteff\ mtCYTB)}$ and $2^{(Cteff\ nUCP1\ -\ Cteff\ mtCYTB)}$. The values were multiplied by 2, to account for two copies of each nuclear gene, and averaged to obtain final mtDNA/nDNA ratios.

**Pig model**. To investigate the potential effect of mode of sampling of liver tissue (percutaneous vs. wedged biopsy) on HRR measurements we sampled three pairs of percutaneous and wedged biopsy liver tissue samples from one pig. The pig was a female Danish Landrace pig, age 18 weeks, weight 51 kg. Tissue was sampled immediately after sacrificing of the animal. The pig had been used for another scientific purpose and all necessary approvals had been obtained (license number 2018-15-0201-01608). All animal experiments were performed in accordance with the Danish law for the protection of animals and the investigation conformed to the guidelines from Directive 2010/63/EU of the European Parliament on the protection of laboratory animals and to the ARRIVE-guidelines. The study had been reviewed and approved by the regional Animal Welfare Inspectorate in Denmark.

**Calculations**. P/E was calculated as OXPHOS$_{max}$(GMOS$_D$)/state u(FCCP). The P/E expresses how close the measured OXPHOS$_{max}$ is to the maximal electron transport system capacity (ETS) in the uncoupled state.

Respiratory control ratio (RCR) was calculated as GM$_D$/GM. The RCR was used as a general measure of mitochondrial function, i.e., the ability of the mitochondria to increase respiration in response to ADP supplementation.

The Homoeostatic Model Assessment of Insulin Resistance (HOMA-IR) was calculated according to the formula [fasting glucose (mg/dL) × insulin (mU/L)/405][54].

The percentage excess body weight loss (%EBWL) at 12 months was calculated as [((baseline BMI – follow-up BMI)/(baseline BMI-25)) × 100%].

**Statistical analyses**. The criteria for normal distribution were not met. Comparisons between the four groups were made using the Kruskal–Wallis test with adjustment for multiple comparison or the Wilcoxon-signed rank test for paired data (baseline vs. 12 months follow-up).

A simple linear regression was used to explore eight predictors; these were selected by the author group based on presumed clinical importance for mitochondrial respiration (OXPHOS$_{max}$) at baseline (liver tissue: BMI, HOMA-IR, steatosis grade, ballooning grade, VAT OXPHOS$_{max}$, plasma leptin, plasma adiponectin and ALT. VAT/SAT: BMI, HOMA-IR, plasma leptin, plasma adiponectin, HbA1c, plasma sCD163, plasma sCD206, VAT/SAT macrophage count. To correct for multiple testing in the regression analyses, the Bonferroni correction was used and a P-value of 0.05/8 = 0.006 was considered significant. The

coefficient estimates are given with 95% confidence intervals (CI). The four most significant variables for each tissue were tried in multiple linear regression models with forward selection and model control, including collinearity testing and log transformation when necessary. Simple logistic regression was used to explore if VAT and SAT adipose tissue OXPHOS influenced the presence of NASH. In post hoc analyses, sex and type 2 diabetes status were tried alongside the four significant variables in each of the three tissues at baseline. Furthermore, in simple regression analysis total weight loss, delta NAS and type of surgery were tried as predictors for delta respiration per mtDNA in liver tissue.

Study participant characteristics are presented as medians with an interquartile range (IQR) or as frequencies. $P < 0.05$ (two-sided) is considered statistically significant. All statistical analyses were performed using IBM SPSS Statistics 25 64-bit.

**Reporting summary**. Further information on research design is available in the Nature Research Reporting Summary linked to this article.

## Data availability

Source Data to figures are provided with this paper. Other data that support the findings of this study are available upon reasonable request from the corresponding author [Julie Steen Pedersen, contact information: julie.steen.pedersen@regionh.dk] on the condition that individual approval can be provided by the Danish Data Protection Agency. The timeframe for response to requests is dependent on the case processing time at the Danish Data Protection Agency. Some data are not publicly available due to restrictions set by the Danish Data Protection Agency as public sharing and sharing of large sets of metadata compromise research participant privacy/consent and hence could violate our Data Agreement with the Danish Data Protection Agency.

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

## Acknowledgements

We would like to thank Christine Rasmussen (Department of Clinical Biochemistry, Copenhagen University Hospital Rigshospitalet), Regitze Kraunsøe and Jeppe Bach (Department of Biomedical Sciences, Faculty of Health and Medical Sciences, University of Copenhagen) for skilful laboratory assistance, project nurse Karen Lisa Hilsted (Gastro Unit, Medical Division) for logistical and administrative assistance. Also, we wish to thank the MicroBLiver Research Consortium. Novo Nordisk Foundation, Challenge Grant "MicroBLiver" (unrestricted grant), grant number NNF15OC0016692: J.S.P., M.O.R., F.B., T.H. Michaelsen Foundation, grant number 10-100012: J.S.P. Aase and Ejnar Danielsen Foundation, grant number 19-10-0284: J.S.P. Novo Nordisk Foundation Center for Protein Research is supported financially by the Novo Nordisk Foundation (Grant agreement NNF14CC0001): N.J.W.A.

## Author contributions

Conceptualisation—ideas; formulation or evolution of overarching research goals and aims: J.S.P., F.B., F.D., S.L., L.L.G., S.M. Data curation—management activities to annotate (produce metadata), scrub data and maintain research data (including software code where necessary for interpreting the data itself) for initial use and later re-use: J.S.P., M.O.R., K.C., S.L., E.G.S., N.J.W.A., A.L.B. Formal analysis—application of statistical, mathematical, computational, or other formal techniques to analyse or synthesise study data: J.S.P., S.L., E.G.S., A.L.B. Funding acquisition—acquisition of the financial support for the project leading to this publication: J.S.P., T.H., F.B., F.D., Z.G.H., N.J.W.A. Investigation—conducting the research and investigation, specifically performing the experiments, or data/evidence collection: J.S.P., M.O.R., K.C., S.L., N.J.W.A., E.G.S., Z.G.H., V.B.K., A.B.B., B.H.O., R.R.S., A.L.B. Methodology—development or design of methodology; creation of models: J.S.P., F.D., S.L., F.B. Project administration—management and coordination responsibility for the research activity planning and execution: J.S.P., M.O.R., K.C., S.L. Resources—provision of study materials, reagents, materials, patients, laboratory samples, animals, instrumentation, computing resources, or other analysis tools: N.J.W.A., E.G.S., Z.G.H., N.J.W.A., V.B.K., A.B.B., B.H.O., S.L., F.D., J.S.P., F.B., B.H.O., R.R.S., A.L.B. Supervision—oversight and leadership responsibility for the research activity planning and execution, including mentorship external to the core team: F.D., F.B., L.L.G., S.M., T.H., S.L. Validation—verification, whether as part of the activity or separate, of the overall replication/reproducibility of results/experiments and other research outputs: S.L., F.D. Visualisation—preparation, creation and/or presentation of the published work, specifically visualisation/data presentation: J.S.P., F.D. Writing—original draft—preparation, creation and/or presentation of the published work, specifically writing the initial draft (including substantive translation): J.S.P., F.D., F.B., S.L. Writing—review & editing—preparation, creation and/or presentation of the published work by those from the original research group, specifically the critical review, commentary or revision—including pre- or post-publication stages: J.S.P., M.O.R., F.D., F.B., S.L., S.M., N.J.W.A., L.L.G., E.G.S., T.H., R.R.S., A.L.B.

## Competing interests

J.S.P., M.O.R., T.H., N.J.W.A. and F.B. receive grants from the Novo Nordisk Foundation. S.M. and L.L.G. consult for and receive grants from Novo Nordisk. The remaining authors declare no competing interests.
