## [Peer Review File · Nature Communications]

REVIEWER COMMENTS

Reviewer #1 (Remarks to the Author):

Pedersen and coauthors investigated mitochondrial oxidative phosphorylation in subjects with and without NAFLD/NASH before and 12 months after bariatric surgery using high-resolution respirometry (HRR). Despite the use of HRR as an elegant way to analyze mitochondrial function, the study does not explore a possible mechanism of action and some results are contradictory. I would encourage additional experiments to explain which is the mechanism driving the beneficial effects on mitochondria observed after bariatric surgery. Moreover, the study population at baseline was composed of 62 subjects undergoing bariatric surgery but only 21 subjects were restudied at 12 months after bariatric surgery. Eventually, data relative to this small number of subjects included in the follow up was jeopardized by a variety of hepatic histological pictures (13 subjects without NAFLD at baseline, 6 subjects with NAFLD and 2 subjects with NASH). Nevertheless, the patients were included in a single group.

MAJOR COMMENTS

The authors hypothesize that OXPHOS patterns in VAT and SAT may provide a mechanistic link to the different NAFLD phenotypes, in which way?.

Indeed, they report that "In regression analyses, none of the chosen eight predictors (BMI, HOMA-IR, steatosis grade, ballooning grade, VAT OXPHOS_{max}, plasma leptin, plasma adiponectin and alanine aminotransferase (ALT)) were found to be significantly associated with liver OXPHOS_{max} per CS".

These results disprove their theory; in fact, VAT OXPHOS did not correlate with liver OXPHOS.

The authors report "The third major, and novel, finding is that complex II and IV respiration, as well as ETS, increased markedly 12 months after surgery (Fig. 5), irrespective of both mode of surgery, NAFLD status at baseline and degree of weight loss (data not shown)". Since liver histology improved in all patients operated of bariatric surgery and the authors exclude a direct effect of weight loss or of surgery itself, which is the mechanism of liver histology improvement?

Although the definition of NASH as the presence of a NAFLD Activity Score (NAS) ≥ 5 has been widely adopted, its use outside the setting of interventional studies was questioned (EM Brunt, Hepatology 2011). Indeed, in her latest study Brunt et al. further highlights that not all biopsies with $NAS \geq 5$ have findings that meet the diagnostic criteria for NASH, while some cases of $NAS \leq 4$ do, indicating that a threshold value of a $NAS \geq 5$ cannot be used reliably to establish the presence or absence of NASH. Please change accordingly.

To this reviewer it is unclear why the authors did not find a significant difference in the hepatic mitochondrial content, especially between lean controls and the three groups with different stages of liver disease. Indeed, changes in hepatic mitochondrial content, evaluated by hepatic citrate synthase activity, precede NAFLD development in both rodents (RS Rector, J Hep. 2010) and human with NAFLD (C Koliaki, Cell Metab. 2015). Please comment.

Another issue regards lacking of data on mitochondrial biogenesis and mitophagy. Liver tissues from patients with early-stage NAFLD have higher mitochondrial biogenesis and mitochondrial mass than liver tissues from healthy individuals. Moreover, NAFLD and NASH are characterized by reduced expression of genes that regulate mitophagy. I suggest to gain further insight on this topic to strengthen the

manuscript.

Since increased respiratory rates do not necessarily reflect efficient electron transport chain, I would encourage the assessment of ATP production. Moreover, elevated ROS, likely derived from uncoupled mitochondria, can promote lipid peroxidation, which potentiates cellular damage. I would like to see some data on this issue.

How do the authors explain the lack of difference in O₂ flux between NASH and control groups reported in figure 2, panels D and F?

Page 11 lines 202-204, the authors state: "Although mtDNA copy number were similar among groups when correcting fluxes for mtDNA content, respiratory fluxes were no longer significantly different".

How do the authors explain this difference?

As stated above the subjects included in the 12 months follow up group are not equally distributed among histological classes. On which base the authors state (page 12 lines 242-243) that NAFLD status at baseline did not influence the increase in mitochondrial respiration. Please comment.

Can the authors explain why they excluded 41 patients at 12 months follow up and why they reorganized the subjects in only one group called "all OBE"?

In the conclusion section (page 17, lines 350-352) the authors state that "weight loss results in improved hepatic mitochondrial respiration", while at page 12, lines 243-244 they affirm that "%EWL did not significantly associate with increased respiration". Please explain.

I would also suggest using the total weight loss instead of %EWL to perform regression analysis.

It would be interesting to see if the increased mitochondrial respiration observed 12 months after bariatric surgery (Figure 5) is similar to the value found in lean controls. Please provide data on this.

Can the authors explain why the data on mitochondrial respiration were not performed in the adipose tissue at 12 months follow up?

MINOR COMMENTS

Please report 3 figures in each P value instead of <0.05

The authors state that "To correct for multiple testing, the Bonferroni correction was used and a p-value of $0.05/8 = 0.006$ was considered significant". However, many values are reported as significant with $P < 0.05$. Please explain.

During surgery, "Tissues were immediately divided into smaller pieces of 50-100 mg, placed in separate tubes containing a chilled mitochondrial preservation buffer (BIOPS)". How did the authors manage to perform all the experiments with the very small sample obtained at the follow-up with the percutaneous liver needle biopsy?

Please do not use obese as an adjective in order to avoid obesity stigma.

Reviewer #2 (Remarks to the Author):

This manuscript aims to determine the role of mitochondrial oxidative phosphorylation (OXPHOS) in the liver, VAT and SAT on NAFLD and NASH. A total of 71 subjects were enrolled in the study: 62 obese individuals undergoing a bariatric surgery and 9 non-obese controls who underwent a cholecystectomy for gallstone disease. Each of these subjects underwent a hepatic, VAT and SAT biopsy procedure at the time of surgery. 21 of the obese subjects who underwent bariatric surgery had an ultrasound-guided liver biopsy 12 months after surgery. Tissue OXPHOS was assessed using Oroboros high-resolution respirators with two different protocols to measure the various mitochondrial respiration parameters of interest. The authors reached 3 conclusions: 1) hepatic OXPHOS is not changed with NASH; 2) VAT and SAT OXPHOS are significantly reduced in NAFLD groups; and 3) mitochondrial respiratory capacity is significantly elevated 12 months after bariatric surgery.

Critique:

This very well written manuscript addresses the very important and unresolved question on the role of liver and adipose tissue mitochondrial function vs. NAFLD/NASH. It reports to my knowledge the largest dataset so far in humans looking at functional ex vivo mitochondrial function at the liver and VAT in patients with NAFLD/NASH. In addition, it applied rigorous methods using ex vivo OXPHOS measurements, with correction using either citrate synthase levels or mitochondrial DNA content. It also used standard pathological definitions of NAFLD and NASH. Statistical analyses are rigorous and results well interpreted. Finally, I find the discussion relevant and complete.

The following comments can hopefully improve this excellent manuscript:

- 1- In view of the importance of age and sex as factors influencing energy metabolism, it is difficult to understand why these were not selected among the important co-variables in the analytic plan. This omission needs to be explained and probably mentioned as a weakness of the study.
- 2- Along the same line, surprisingly diabetes status was not included among the 8 most likely factors to influence hepatic OXPHOS. However, diabetes status was later examined in the text, without showing the data, and ruled out as a significant factor to explain the cohorts' hepatic OXPHOS variance. The omission of T2D among the initial factors need to be addressed. Furthermore, the posthoc analysis on T2D probably need to be integrated in supplements and better explained.
- 3- Line 164 and following: review the text to avoid giving the impression that there is a difference between NAFLD+ vs. other groups in OXPHOS.
- 4- Throughout the results and discussion, the authors refer to reduced OXPHOS in SAT and VAT among NAFLD groups vs. control. However, this is driven as much by NAFLD- obese subjects than the other two groups. Therefore, it would be more correct to refer to obese groups, notwithstanding NAFLD.
- 5- With regards to the increase in maximal uncoupled hepatic OXPHOS after surgery, the authors refer

to hepatic OXPHOS capacity. It is however unclear what is the in vivo physiological significance of ex vivo OXPHOS under such extreme conditions. This would need some acknowledgement in the discussion.

6- Line 242: it is mentioned that the type of surgery or NAFLD status at baseline had no influence on the OXPHOS outcome. It would be informative to give the data in the supplements.

Minor comments:

1- Please standardize the use of NAFL vs. NAFLD

2- Remove 'majorly' on line 116.

3- On line 158, indicate that values are median (IQR) at first appearance.

4- Line 171: add 'being' before 'numerically'.

5- Line 228: add 'the' before 'presence'.

Reviewer #3 (Remarks to the Author):

This manuscript describes the results of a clinical study to determine whether impaired mitochondrial oxidative phosphorylation (OXPHOS) in liver tissue, visceral (VAT) and subcutaneous adipose tissue (SAT) are associated with NAFLD severity and how hepatic OXPHOS responds to improvement in NAFLD.

This is a well conducted study providing a valuable OXPHOS activity in 2 tissues (liver and VAT) that are not often studied and also in SAT. In addition to cross sectional results across severity of NAFLD, longitudinal data following bariatric surgery is also presented in a smaller number of participants.

Interestingly, liver mitochondrial respiration was not related to NAFLD severity (control through NASH). There were few statistically significant differences between groups, that were inconsistent across conditions (Suit P1 vs P2, unadjusted, adjusted for CS, adjusted for mtDNA). Based on these observations, the discussion (lines 258-263) regarding "downward slopes" and "higher numerical fluxes" and "in general have higher coupled and uncoupled oxygen fluxes" should be eliminated.

Unlike liver mitochondrial respiration, adipose tissue respiration, particularly SAT, was significantly lower in those with NAFL and NASH compared to control participants.

Although the longitudinal aspect of this study had the potential to be very informative, biopsy procedures and lack of quantification of mitochondrial content reduces enthusiasm. There was a significant increase in mitochondrial respiration in response to weight loss induced by bariatric surgery. However, the baseline and post-surgery biopsies were obtained by different procedures (wedge vs percutaneous). Data comparing the 2 procedures in pigs (n=3) showed no significant difference, but it appeared that mitochondrial respiration may have been higher with the wedge procedure. Therefore, the biopsy differences may not be an issue since a higher mtResp was observed post-surgery using the percutaneous procedure. Unfortunately, unlike there were apparently no CS or mtDNA measures to determine if this increased mtResp is due to increased mitochondrial content or intrinsic mtResp. Furthermore, the clinical relevance of this increased liver mtResp is in response to bariatric surgery since

liver mtResp was not lower in NAFL or NASH at baseline. Nonetheless this is an interesting finding. Some discussion of this is needed.

The other major deficiency of the longitudinal study was the lack of adipose tissue mtResp, particularly in SAT in which the most robust difference in mtResp was observed compared to CON.

The discussion related to “differences” or “no differences” in those with T2DM should be eliminated, since the sample included only a very small sample of individuals with T2DM.

Point-by-Point response

Reviewer #1 (Remarks to the Author):

Pedersen and coauthors investigated mitochondrial oxidative phosphorylation in subjects with and without NAFLD/NASH before and 12 months after bariatric surgery using high-resolution respirometry (HRR). Despite the use of HRR as an elegant way to analyze mitochondrial function, the study does not explore a possible mechanism of action and some results are contradictory. I would encourage additional experiments to explain which is the mechanism driving the beneficial effects on mitochondria observed after bariatric surgery. Moreover, the study population at baseline was composed of 62 subjects undergoing bariatric surgery but only 21 subjects were restudied at 12 months after bariatric surgery. Eventually, data relative to this small number of subjects included in the follow up was jeopardized by a variety of hepatic histological pictures (13 subjects without NAFLD at baseline, 6 subjects with NAFLD and 2 subjects with NASH). Nevertheless, the patients were included in a single group.

We thank reviewer number 1 for the valuable comments which have helped us to improve our manuscript. In the prospective part of the study we have now performed new analyses in bio banked liver tissue sampled at 12 months after surgery and added data on mtDNA/nDNA in the liver tissue 12 months after surgery.

Regarding the 21 subjects re-studied at 12 months it is more thoroughly discussed during question 11 and 12.

MAJOR COMMENTS

1) The authors hypothesize that OXPHOS patterns in VAT and SAT may provide a mechanistic link to the different NAFLD phenotypes, in which way?

Answer: Thank you for your question. Previously it has been shown that the degree of adipose tissue inflammation is correlated with NAFLD severity (e.g. Canello et al. Diabetes 2006).

SAT and VAT do not present with the same mitochondrial respiratory capacity, as VAT has a lesser respiratory reserve capacity compared with SAT (Kraunsøe et al, Am J Phys, 2010). Thus, a decreased capacity to oxidize substrates in VAT may indirectly negatively influence the liver, because of potential lipid peroxidation, and cell damage and subsequent release of inflammatory cytokines/adipokines to the liver via the portal vein. In addition, a diminished capacity to oxidize lipids may result in increased release of FFA into the portal vein, thereby facilitating hepatic steatosis.

Change in MS: We have now expanded this notion in the background section (line 144-149).

2) Indeed, they report that "In regression analyses, none of the chosen eight predictors (BMI, HOMA-IR, steatosis grade, ballooning grade, VAT OXPHOSmax, plasma leptin, plasma adiponectin and alanine aminotransferase (ALT)) were found to be significantly associated with liver OXPHOSmax per CS". These results disprove their theory; in fact, VAT OXPHOS did not correlate with liver OXPHOS.

Answer. Yes, that is true. And as already mentioned in our conclusion we state that 'compromised AT respiration is not likely to significantly impact NAFLD severity'.

We hypothesized that we would find significantly decreased OXPHOS in adipose tissue – at least in those with NASH. We found no such evidence for our hypothesis. The fact that adipose tissue OXPHOS did not correlate to liver OXPHOS (neither when expressed as OXPHOS per CS or OXPHOS per mtDNA) is already mentioned in line 201-203. Also, in logistic regression neither VAT nor SAT OXPHOS could predict NASH yes/no (already mentioned line 245-246).

Change in MS: We have now further commented on this finding in the discussion (discussion line 336-337 and 357-358).

3) The authors report “The third major, and novel, finding is that complex II and IV respiration, as well as ETS, increased markedly 12 months after surgery (Fig. 5), irrespective of both mode of surgery, NAFLD status at baseline and degree of weight loss (data not shown)”. Since liver histology improved in all patients operated of bariatric surgery and the authors exclude a direct effect of weight loss or of surgery itself, which is the mechanism of liver histology improvement?

Answer: Thank you for your question. The reviewer addresses a question (mechanism of histological improvement), which has been asked by researchers without definitive answer for decades.

We cannot from our data conclude, that there is a relation between the observed increase in respiration and mtDNA/nDNA with the improvement in histology. In regression analyses neither degree of weight loss, type of surgery nor decrease in NAS were significantly associated to neither increase in respiration. Of note (and not part of this study) we have 40 patients with new histological assessment 12 mo after surgery and in this dataset we have not been able to conclude on the predictors for NAFLD improvement (weight loss, decrease in HOMA—IR etc).

In the present study a substantial weight loss was observed in all study subjects and a substantial degree of NAFLD improvement was also observed in all individuals pointing towards the same beneficial effect of bariatric surgery in all study subject. The lack of association between weight loss (delta BMI/delta EBWL/Total weight loss) and histological improvement can therefore partly be explained by the uniformity of the data (the same improvement in all) and to the rather low number of patients in the 12 months follow-up group.

We still believe that the weight loss must play a significant role in the observed changes in both respiration and histological improvement, though we from the present data cannot conclude this. To our knowledge it has never been proven whether it is the weight loss itself or the many beneficial side effects of weight loss (which each are inter-connected) – e.g. increased peripheral and hepatic insulin sensitivity, restoration of adipokine imbalance and decreased tissue and systemic inflammation. We believe that it is combination of a multitude of the factors mentioned.

Change in MS: Based on your comment we have in general ‘sharpened’ our statements (and deleted some lines e.g. Abstract line 105, Discussion line 374-375) to make sure that we refer to what we can deduce from *our* data. We have added details on the analyses in Results line 267-273.

We have also specified that we from our data cannot exclude an effect of the mentioned parameters included e.g. Discussion line 388-393.

4) Although the definition of NASH as the presence of a NAFLD Activity Score (NAS) ≥ 5 has been widely adopted, its use outside the setting of interventional studies was questioned (EM Brunt, Hepatology 2011). Indeed, in her latest study Brunt et al. further highlights that not all biopsies with NAS ≥ 5 have findings that meet the diagnostic criteria for NASH, while some cases of NAS ≤ 4 do, indicating that a threshold value of a NAS ≥ 5 cannot be used reliably to establish the presence or absence of NASH. Please change accordingly.

Answer: Application of the various histological scoring systems to approximate ‘true’ NAFLD severity is a subject of much debate. We acknowledge the shortcomings of the NAS. NAS was chosen as this is/was the most widely used staging in previous studies addressing the same subject – e.g. the study by Koliaki et al.

We have tried to present the data, so that it would be comparable with the Koliaki study. Should we change accordingly (e.g. to the FLIP algorithm) the result would be two groups (NASH and no-NASH) and the current NASH group and the current NAFL+ group would be grouped together. This was not the intention of our study as we wanted to explore the spectrum of NAFLD. We are willing to do the analyses if requested but we believe this would lead to great loss of information.

Change in MS: None

5) To this reviewer it is unclear why the authors did not find a significant difference in the hepatic mitochondrial content, especially between lean controls and the three groups with different stages of liver disease. Indeed, changes in hepatic mitochondrial content, evaluated by hepatic citrate synthase activity, precede NAFLD development in both rodents (RS Rector, J Hep. 2010) and human with NAFLD (C Koliaki, Cell Metab. 2015). Please comment.

Answer: Thank you for the question. We were also surprised to see the lack of difference in mtDNA content between groups – especially in the liver. Therefore, we decided to evaluate CSA in liver tissue to evaluate if results were explained by the applied methodology. Again, we found no difference in CSA between groups in liver tissue. We are confident in our methods and the analytical approach. There could be several explanations for the lack of difference. However, we are puzzled by the fact that Koliaki et al measure a 50% increased mitochondrial mass in their NASH study subjects. We cannot reproduce their mitochondrial mass data in our human dataset. In one of our previous studies we also did not find a difference in CSA between lean CONs, individuals with obesity and individuals with obesity+type2 diabetes (*Hepatic mitochondrial oxidative phosphorylation is normal in obese patients with and without type 2 diabetes, Lund et. al. J. Physiol.2016*)

In addition, we find that the study by Rector RS et al. contradicts the findings of Koliaki et al, since CSA started to decrease from week 4 in their study with a significant decrease in CSA at 40 weeks in OLETF rats (when the rats had developed NASH) compared to lean rats.

Change in MS: None

6) Another issue regards lacking of data on mitochondrial biogenesis and mitophagy. Liver tissues from patients with early-stage NAFLD have higher mitochondrial biogenesis and mitochondrial mass than liver tissues from healthy individuals. Moreover, NAFLD and NASH are characterized by reduced expression of genes that regulate mitophagy. I suggest to gain further insight on this topic to strengthen the manuscript.

Answer: We agree that this would have been of interest. However, we do not have data on hepatic/adipose tissue mitochondrial genes. However, we have added new analyses on the mtDNA/nDNA content 12 months after surgery and indeed we see a significant increase in mtDNA/nDNA from baseline to 12 months after surgery which could point towards increased mitochondrial biogenesis.

Change in MS: Abstract Line 105, Results line 260-264, Discussion line 372-373, 379-382, 398-402 and conclusion 449-450

7) Since increased respiratory rates do not necessary reflect efficient electron transport chain, I would encourage the assessment of ATP production.

Answer: It is correct that respiration does not necessarily reflect electron transport chain efficiency but the FCCP step is a measure of maximal electron transport chain capacity and hence an indirect measure of performance of the electron transport chain.

We did not measure ATP production in the fresh tissue and consequently we do not have the possibility to add the requested data post hoc.

Measurement of ATP production furthermore requires a tissue biopsy of a size that allows isolation of the mitochondria. This procedure does, however, not secure that all mitochondria are included. In fact, the yield can be so low that the data may not represent the “true” value. Finally, the size of 12 mo biopsies did not allow for this procedure.

8) Moreover, elevated ROS, likely derived from uncoupled mitochondria, can promote lipid peroxidation, which potentiates cellular damage. I would like to see some data on this issue.

Answer: Though preferable we are not able to provide data on this. We had to limit ourselves with the tissue material that was obtained, and we focused on measurement that could be performed in both the wedge and the needle biopsy specimens. ROS (or to be accurate, superoxide) can be measured, but it requires separate analyses in a separate SUIT protocol.

Change in MS: None

9) How do the authors explain the lack of difference in O₂ flux between NASH and control groups reported in figure 2, panels D and F

Answer: We are aware of the lack of difference. Please refer to question/answer number 5.

Change in MS: None

10) Page 11 lines 202-204, the authors state: “Although mtDNA copy number were similar among groups when correcting fluxes for mtDNA content, respiratory fluxes were no longer significantly different”. How do the authors explain this difference?

Answer: The disappearance of the statistical significance is probably ascribed the small numerical differences in mtDNA/nDNA content and CSA content between groups. Please refer to question/answer 5.

Change in MS: None

11) As stated above the subjects included in the 12 months follow up group are not equally distributed among histological classes. On which base the authors state (page 12 lines 242-243) that NAFLD status at baseline did not influence the increase in mitochondrial respiration. Please comment.

Answer: Thank you for the clarifying question. When we compared respiratory rates at 12 months between the NAFLD groups based on the histological status at baseline, we found no difference in neither flux rates at 12 months nor delta changes between baseline and 12 months

between the NAFLD groups (NAFL-, NAFL+, NASH). As mentioned by yourself earlier we had a low number of study subjects in each group at 12 months when stratified by baseline histology.

We agree that the study is underpowered to fully answer this question and we have erased this from the manuscript.

In addition, we have now added the data on delta NAS as a predictor for delta respiration per mtDNA 12 months after surgery: No statistically significant association between these two variables were found (see results + supplementary material).

Based on your comment we have now added that though we found no association between e.g. change in NAS and respiration at 12 months we cannot exclude an effect of these parameters due to the rather low numbers of patients included in the study, which hampers a subgroup analysis.

Change in MS:

Erased line 373-374.

Results line 267-271, Discussion line 367-371.

12) Can the authors explain why they excluded 41 patients at 12 months follow up and why they reorganized the subjects in only one group called "all OBE"?

Answer: This was a matter of study design prior to initiation of the study. Ethically we did not want to re-biopsy patients who had a normal liver at baseline. Consequently, we designed the 12-months follow-up protocol to only include those patients who had signs of activity or fibrosis at baseline and exclude the rest (though only two had completely 'healthy' livers at baseline). Secondly, we made it a matter of choice for the patients to undergo a second biopsy at 12 months. Some declined. Others fell for other exclusion criteria though they had initially consented to the repeated liver biopsy (e.g. withdrew consent, anxiety at day of biopsy etc.). Also, we only included patients who had completely paired P1 HRR runs at both baseline and 12 months. That meant that we excluded 3 patients who fell for the data quality check (too high cytochrome c response at 12 months leaving us with 21 subjects for follow-up. We have included a flowchart in the MS for clarification (supplementary figure 2). Finally, because this study was part of a larger study we had to set a deadline for conclusion of HRR analyses at 12 months.

We chose to organize the subjects into one group (all OBE) when comparing baseline respiration with 12 months respiration because the individual groups (NAFL-, NAFL+, NASH) would become too small for statistical analysis. We feel rather confident to do this, since we found no significant respiratory difference between any of the NAFLD groups and CONs at baseline.

At baseline the 21 study subjects available for 12 months analysis did not significantly separate from the 41 who did not conclude in the 12 months analyses in terms of anthropometrics, phenotype, biochemical profile, NAFLD activity/fibrosis scores, fluxes, mtDNA/nDNA or CSA.

Change in MS: Addition of supplementary figure S2 + line 560-562

13) In the conclusion section (page 17, lines 350-352) the authors state that "weight loss results in improved hepatic mitochondrial respiration", while at page 12, lines 243-244 they affirm that "%EWL did not significantly associate with increased respiration". Please explain.

I would also suggest using the total weight loss instead of %EWL to perform regression analysis.

Answer: Thank you. It is correct that we from our data cannot conclude whether it was the weight loss per se that resulted in increased respiration. We have changed accordingly in the abstract line 105 and again modified our statement in the final conclusion line 450.

We chose %EBWL because this measure is more independent of preoperative BMI/weight and therefore is often regarded as a more meaningful/unbiased measure of weight loss but we have now added total weight loss to the table 3 and analyses.

Expressing weight loss as total weight loss did not alter our results of a non-significant relationship between degree of weight loss and OXPHOS results at 12 months and this has been added to the manuscript (Line 267-269, Discussion: 388-393).

Change in MS: Please see the changes above

It would be interesting to see if the increased mitochondrial respiration observed 12 months after bariatric surgery (Figure 5) is similar to the value found in lean controls. Please provide data on this.

Answer: Thank you for the suggestion. We agree.

Change in MS: We have changed the figures accordingly so that CON (grey/dimmed) is now a part of figure 5 and 6 at 12 months for reference. The 21 individuals with obesity presented with numerically higher fluxes 12 months after surgery compared with CON, but this was not significant.

Figures displaying flux/mtDNA at 12 months have also been added to the manuscript (Fig 5+6)

14) Can the authors explain why the data on mitochondrial respiration were not performed in the adipose tissue at 12 months follow up?

Answer: Though this would have been of high interest, we unfortunately found it impossible to sample omental tissue (VAT) non-invasively and we could not ethically defend the necessity for performing re-laparoscopy in general anesthesia only for omental adipose tissue sampling purposes in a research setting. We *did* investigate the possibility of sampling omental tissue ultrasonically guided, but our interventional radiologists said that this was not practically feasible. We debated several times, whether we should sample SAT 12 months postoperatively, but this would imply excision of tissue in local anesthesia and as we were mostly interested in VAT, we consequently refrained from sampling any adipose tissue at follow-up. In retrospect (we did not evaluate any of the data before end-of-study) we could have wished that we sampled SAT, which to our surprise appeared to be more affected than VAT.

Change in MS: None

MINOR COMMENTS

15) Please report 3 figures in each P value instead of <0.05

Answer and change in MS: We have now provided the specific p-values for p<0.05. For p<0.01 and p<0.001 in all figures.

16) The authors state that "To correct for multiple testing, the Bonferroni correction was used and a p-value of $0.05/8 = 0.006$ was considered significant". However, many values are reported as significant with P<0.05. Please explain.

Answer: Only when reporting the results from the linear regressions we set the significance level to 0.006. Otherwise the significance level was set to 0.05. This is stated in the statistical methods section line 628-629.

Change in MS: None

17) During surgery, "Tissues were immediately divided into smaller pieces of 50-100 mg, placed in separate tubes containing a chilled mitochondrial preservation buffer (BIOPS)". How did the authors manage to perform all the experiments with the very small sample obtained at the follow-up with the percutaneous liver needle biopsy?

Answer: Thank you for the correction. In the manuscript we have now clarified further that the division of the tissue in the smaller pieces of 50-100 mg only took place at baseline (baseline biopsy). At 12 months the entire percutaneous sample was placed in buffer and subsequently transported to the laboratory to undergo HRR. The biopsies at 12 months were *not* divided as stated in the manuscript. This has been corrected.

Change in MS: Also, it has been added that in total *three* pieces of tissue were sampled at 12 months per patient; one piece for HRR, one piece for histology and one piece frozen for the biobank. This piece has now been used for mtDNA analysis at 12 months.

Methods line 485 + 487 and line 491-492

18) Please do not use obese as an adjective in order to avoid obesity stigma.

Answer: Thank you for this notion.

Change in MS: We have changed accordingly throughout the manuscript (not highlighted in yellow, but all changes are there) but have remained the abbreviation OBE.

Reviewer #2 (Remarks to the Author):

This manuscript aims to determine the role of mitochondrial oxidative phosphorylation (OXPHOS) in the liver, VAT and SAT on NAFLD and NASH. A total of 71 subjects were enrolled in the study: 62 obese individuals undergoing a bariatric surgery and 9 non-obese controls who underwent a cholecystectomy for gallstone disease. Each of these subjects underwent a hepatic, VAT and SAT biopsy procedure at the time of surgery. 21 of the obese subjects who underwent bariatric surgery had an ultrasound-guided liver biopsy 12 months after surgery. Tissue OXPHOS was assessed using Oroboros high-resolution respirators with two different protocols to measure the various mitochondrial respiration parameters of interest. The authors reached 3 conclusions: 1) hepatic OXPHOS is not changed with NASH; 2) VAT and SAT OXPHOS are significantly reduced in NAFLD groups; and 3) mitochondrial respiratory capacity is significantly elevated 12 months after bariatric surgery.

Critique:

This very well written manuscript addresses the very important and unresolved question on the role of liver and adipose tissue mitochondrial function vs. NAFLD/NASH. It reports to my knowledge the largest dataset

so far in humans looking at functional ex vivo mitochondrial function at the liver and VAT in patients with NAFLD/NASH. In addition, it applied rigorous methods using ex vivo OXPHOS measurements, with correction using either citrate synthase levels or mitochondrial DNA content. It also used standard pathological definitions of NAFLD and NASH. Statistical analyses are rigorous and results well interpreted. Finally, I find the discussion relevant and complete.

We thank reviewer 2 for the very positive and constructive comments. We are pleased that this reviewer finds our human data well interpreted and our results and manuscript interesting.

The following comments can hopefully improve this excellent manuscript:

- 1- In view of the importance of age and sex as factors influencing energy metabolism, it is difficult to understand why these were not selected among the important co-variates in the analytic plan. This omission needs to be explained and probably mentioned as a weakness of the study.
- 2- Along the same line, surprisingly diabetes status was not included among the 8 most likely factors to influence hepatic OXPHOS. However, diabetes status was later examined in the text, without showing the data, and ruled out as a significant factor to explain the cohorts' hepatic OXPHOS variance. The omission of T2D among the initial factors need to be addressed. Furthermore, the posthoc analysis on T2D probably need to be integrated in supplements and better explained.

Answer to comment 1 + 2: Thank you for your comments. We agree with your observations. We did an initial screening for difference between T2DM (yes/no) and sex (male/female) and found no difference in OXPHOS in the three tissues. However, based on your comment we have now done a post hoc analysis (multiple linear regression) with inclusion of age, sex and T2DM in the model with the four variables with the lowest p-value. Neither T2DM, sex nor age were significant in the hepatic OXPHOS model (T2DM, $p=0.258$, sex: $p=0.634$, age: $p=0.214$) and did not influence the result of BMI being the only significant predictor of hepatic OXPHOS.

Change in MS: We have added this information in the results section (line 205-207 and 243-245 and also mentioned in the post hoc analysis under 'statistics' line 633-636.

Details of the statistical output with regards to the post hoc analyses can now be found in the supplementary material (results) for both liver tissue, VAT and SAT.

- 3- Line 164 and following: review the text to avoid giving the impression that there is a difference between NAFLD+ vs. other groups in OXPHOS.

Answer: Thank you for this notion. We agree.

Change in MS: We have modified accordingly throughout the results section.

- 4- Throughout the results and discussion, the authors refer to reduced OXPHOS in SAT and VAT among NAFLD groups vs. control. However, this is driven as much by NAFLD- obese subjects than the other two groups. Therefore, it would be more correct to refer to obese groups, notwithstanding NAFLD.

Answer: We agree, that obesity seem to be the major determinant of OXPHOS in adipose tissue rather than severity of NAFLD as those with liver steatosis grade 0 (< 5% hepatic triglyceride accumulation) , that being our NAFL- group presented with a similar decrease in OXPHOS as those with NAFL+ and NASH. We have mentioned obesity as the 'common denominator' both in the discussion as well as in the conclusion. Though we understand your comment we are afraid that changing the 'nomenclature' will be confusing and to secure uniformity we have kept the phrase 'NAFLD-subgroups' throughout the manuscript.

Change in MS: None

5- With regards to the increase in maximal uncoupled hepatic OXPHOS after surgery, the authors refer to hepatic OXPHOS capacity. It is however unclear what is the in vivo physiological significance of ex vivo OXPHOS under such extreme conditions. This would need some acknowledgement in the discussion.

Answer: Thank you for your comment.

Change in MS: We have now included this in the discussion line 412-422.

6- Line 242: it is mentioned that the type of surgery or NAFLD status at baseline had no influence on the OXPHOS outcome. It would be informative to give the data in the supplements.

Answer: Yes, we agree. This was also commented by reviewer 1. We have now added this information in the supplements, and we have provided more details throughout the MS.

Change in MS: Results line 267-271, Discussion line 367-371 + supplementary material (results).

Minor comments:

1- Please standardize the use of NAFL vs. NAFLD

Answer: Thank you for your comment. This is a common challenge in most NAFLD papers. We have tried only to use NAFLD when we refer to the total spectrum of the disease (ranging from NAFL, over NASH to NASH-cirrhosis) and to the best of our ability standardized the nomenclature in the manuscript. We have, however, kept the expression 'NAFLD groups' (see your comment/answer no. 4), though we are aware that this phrase also includes NAFL-.

2- Remove 'majorly' on line 116.

Answer and change in MS: This has been done (Note: not highlighted in yellow in MS but *has* been corrected)

3- On line 158, indicate that values are median (IQR) at first appearance.

Answer and change in MS: This has now been added (Note: not highlighted in yellow in MS but *has* been corrected)

4- Line 171: add 'being' before 'numerically'.

Answer and change in MS: Yes, this has now been corrected. (Note: not highlighted in yellow in MS but *has* been corrected)

5- Line 228: add 'the' before 'presence'.

Answer and change in MS: This has now been corrected. (Note: not highlighted in yellow in MS but *has* been corrected)

Reviewer #3 (Remarks to the Author):

This manuscript describes the results of a clinical study to determine whether impaired mitochondrial oxidative phosphorylation (OXPHOS) in liver tissue, visceral (VAT) and subcutaneous adipose tissue (SAT) are associated with NAFLD severity and how hepatic OXPHOS responds to improvement in NAFLD.

This is a well conducted study providing a valuable OXPHOS activity in 2 tissues (liver and VAT) that are not often studied and also in SAT. In addition to cross sectional results across severity of NAFLD, longitudinal data following bariatric surgery is also presented in a smaller number of participants.

We thank reviewer 3 for the very positive and constructive comments. We have performed additional analyses on mtDNA/nDNA content 12 months after surgery to strengthen our longitudinal data substantially (please refer to your comment and our answer no 2).

1) Interestingly, liver mitochondrial respiration was not related to NAFLD severity (control through NASH). There were few statistically significant differences between groups, that were inconsistent across conditions (Suit P1 vs P2, unadjusted, adjusted for CS, adjusted for mtDNA). Based on these observations, the discussion (lines 258-263) regarding "downward slopes" and "higher numerical fluxes" and "in general have higher coupled and uncoupled oxygen fluxes" should be eliminated.

Answer and change in MS: We agree, and we have changed accordingly throughout results and discussion sections.

2) Unlike liver mitochondrial respiration, adipose tissue respiration, particularly SAT, was significantly lower in those with NAFL and NASH compared to control participants.

Although the longitudinal aspect of this study had the potential to be very informative, biopsy procedures and lack of quantification of mitochondrial content reduces enthusiasm. There was a significant increase in mitochondrial respiration in response to weight loss induced by bariatric surgery. However, the baseline and post-surgery biopsies were obtained by different procedures (wedge vs percutaneous). Data comparing the 2 procedures in pigs (n=3) showed no significant difference, but it appeared that mitochondrial respiration may have been higher with the wedge procedure. Therefore, the biopsy differences may not be an issue since a higher mtResp was observed post-surgery using the percutaneous procedure.

Unfortunately, unlike there were apparently no CS or mtDNA measures to determine if this increased mtResp is due to increased mitochondrial content or intrinsic mtResp.

Answer: Thank you for your observations and comments. We agree. We have now performed additional analyses and now provide data on mtDNA/nDNA content in hepatic tissue 12 months after surgery. We show a significant increase by 32% in hepatic mtDNA/nDNA content, which we believe to be the primary driver of the observed increased respiration 12 months after surgery.

Change in MS: Abstract line 105

Results Line 260-265

Discussion line 277-279, 372-373, 379-382, 395-402, 449-450, Methods 491-492.

3) Furthermore, the clinical relevance of this increased liver mtResp is in response to bariatric surgery since liver mtResp was not lower in NAFL or NASH at baseline. Nonetheless this is an interesting finding. Some discussion of this is needed.

Answer: Thank you for your comment. We also find this highly interesting. Please refer to comment/answer no 2. We have now measured mtDNA/nDNA content in liver tissue 12 months after surgery and elaborate on this finding throughout the MS. We find evidence of increased mitochondrial biogenesis 12 months after surgery and this is probably the main driver of the increased respiration.

Change in MS: Abstract line 105

Results Line 260-265

Discussion line 277-279, 372-373, 379-382, 395-402, 449-450, Methods 491-492.

4) The other major deficiency of the longitudinal study was the lack of adipose tissue mtResp, particularly in SAT in which the most robust difference in mtResp was observed compared to CON.

Answer: Thank you for your comment. Though this would have been of major interest, we unfortunately found it impracticable to sample omental tissue non-invasively and we could not ethically defend the necessity for performing re-laparoscopy in general anesthesia only for visceral adipose tissue sampling purposes in a research setting. We *did* investigate the possibility to sample omental tissue ultrasonically guided, but our interventional radiologists said that this was not practically feasible. We debated several times, whether we should sample SAT 12 months postoperatively, but this would imply excision of tissue in local anesthesia and as we were mostly interested in VAT, we consequently refrained from sampling any adipose tissue at follow-up. In retrospect (we did not evaluate any of the data before end-of-study) we could have wished that we *had* sampled SAT, which to our surprise appeared to be more affected than VAT at baseline.

Change in MS: None

5) The discussion related to “differences” or “no differences” in those with T2DM should be eliminated, since the sample included only a very small sample of individuals with T2DM.

Answer: We find your comment relevant. However, reviewer 2, has asked for more details of the influence of T2DM and consequently we have added some extra info on this subject.

Change in MS: Results section (line 205-207 and 243-245 and mentioned in the post hoc analysis under 'statistics' line 633-636).

REVIEWER COMMENTS

Reviewer #1 (Remarks to the Author):

Question 13

The author respond: "Change in MS: We have changed the figures accordingly so that CON (grey/dimmed) is now a part of figure 5 and 6 at 12 months for reference. The 21 individuals with obesity presented with numerically higher fluxes 12 months after surgery compared with CON, but this was not significant. Figures displaying flux/mtDNA at 12 months have also been added to the manuscript (Fig 5+6)"

Indeed, the lack of significance means only that the study was underpowered.

The authors state in the response letter "Of note (and not part of this study) we have 40 patients with new histological assessment 12 mo after surgery and in this dataset we have not been able to conclude on the predictors for NAFLD improvement (weight loss, decrease in HOMA—IR etc)."

I strongly suggest adding this data to the present study.

Answer 14

"We debated several times, whether we should sample SAT 12 months postoperatively, but this would imply excision of tissue in local anesthesia".

The tru-Cut needle biopsy of the adipose tissue biopsy is a common technique performed with minimum discomfort for the patient.

Please add this in the limitations of the study.

Question 16

The authors state "Only when reporting the results from the linear regressions we set the significance level to 0.006. Otherwise the significance level was set to 0.05. This is stated in the statistical methods section line 628- 629".

Indeed this statistical procedure is incorrect, Bonferroni correction should be used also when reporting significances in Table 1.

Question 17

The question was "Tissues were immediately divided into smaller pieces of 50-100 mg, placed in separate tubes containing a chilled mitochondrial preservation buffer (BIOPS)". How did the authors manage to perform all the experiments with the very small sample obtained at the follow-up with the percutaneous liver needle biopsy?

The authors answered: "Thank you for the correction. In the manuscript we have now clarified further that the division of the tissue in the smaller pieces of 50-100 mg only took place at baseline (baseline biopsy). At 12 months the entire percutaneous sample was placed in buffer and subsequently transported to the laboratory to undergo HRR. The biopsies at 12 months were not divided as stated in the manuscript. This has been corrected".

The weight of the individual ultrasound-guided liver specimens is about 17 mg with larger needles, such as 14-gauge core.

Since part of the biopsy was used for histology, how did the authors manage to make the analyses of

this smallest liver samples? Did they refine the technique? In fact, at baseline they used 50-100 mg of liver tissue.

Reviewer #2 (Remarks to the Author):

Thank you for the responses you provided to my previous comments. I find them satisfactory. Furthermore, I appreciate the addition of the mitochondrial/nuclear DNA ratio after bariatric surgery, which strengthens your conclusion.

Here are additional suggested modifications from some of the added parts of the revised version of the manuscript:

Lines 144-149 : Reference 29 demonstrated that VAT display greater oxidative phosphorylation rate than SAT. Therefore, it is unclear why the authors refer to a lower respiratory reserve in VAT.

Line 382: remove 'but' at the end of the first sentence.

Line 385: remove 'increase' before gene.

Line 386: replace 'significantly' by 'significant'.

Line 410: 'of' instead of 'og'.

Line 411: remove 'In this case... mitochondria.'

Line 412: remove 'measure' before mitochondrial.

Line 420: Not clear what the authors mean by 'The resolution is not near the resolution one can obtain with the ex vivo technique'. PET is by far the most sensitive technique there is, with a sensitivity in the range of pico to femtomolar. It may be more accurate to state that in vivo techniques usually provide proxy measures of mitochondrial function such as ATP/creatinine ratio or oxygen consumption.

Reviewer #3 (Remarks to the Author):

The authors have addressed the issues I raised in the previous review. However, the additional discussions regarding T2DM made me realize that I did not see any discussion of T2DM resolution in response to surgery. Did any of the participants see a resolution in their T2DM status, and if so, how many, and were there enough to see if any of the outcomes associated with resolution?

Minor Issues

1. The wording of baseline differences should change from increased/decreased, to higher/lower
2. Line 335: "Secondly, did we not prove any correlation .." should be "we did not"

REVIEWER COMMENTS

Reviewer #1 (Remarks to the Author):

Thank you for your comments to our revised manuscript.

Question 13

The author respond: "Change in MS: We have changed the figures accordingly so that CON (grey/dimmed) is now a part of figure 5 and 6 at 12 months for reference. The 21 individuals with obesity presented with numerically higher fluxes 12 months after surgery compared with CON, but this was not significant. Figures displaying flux/mtDNA at 12 months have also been added to the manuscript (Fig 5+6)"

Indeed, the lack of significance means only that the study was underpowered.

Answer: Yes, we do not argue with this interpretation.

The authors state in the response letter "Of note (and not part of this study) we have 40 patients with new histological assessment 12 mo after surgery and in this dataset we have not been able to conclude on the predictors for NAFLD improvement (weight loss, decrease in HOMA—IR etc)."

I strongly suggest adding this data to the present study.

Answer: This data have already been published in full length in *JCM*, August 2021 'Effects of Roux-en-Y gastric Bypass and Sleeve gastrectomy on Non-Alcoholic Fatty Liver Disease: A 12-Months Follow-Up Study With Paired Liver Biopsies' (DOI:[10.3390/jcm10173783](https://doi.org/10.3390/jcm10173783)) and we have chosen not to re-publish/add this data to the current MS but provided a reference in the discussion (ref 48, line 390).

We have also now provided the specific data on associations between change in NAS and change in weight, HOMA-IR for all 40 patients with follow-up biopsies available (see supplementary material line 81-88).

Answer 14

"We debated several times, whether we should sample SAT 12 months postoperatively, but this would imply excision of tissue in local anesthesia".

The tru-Cut needle biopsy of the adipose tissue biopsy is a common technique performed with minimum discomfort for the patient.

Please add this in the limitations of the study.

Answer: That is correct. This has been added to 'limitations' line 433.

Question 16

The authors state "Only when reporting the results from the linear regressions we set the significance level to 0.006. Otherwise the significance level was set to 0.05. This is stated in the statistical methods section line 628- 629".

Indeed this statistical procedure is incorrect, Bonferroni correction should be used also when reporting significances in Table 1.

Answer: It is not exactly clear to us what is being asked for in this comment/question.

If the reviewer refers to table 1 (baseline characteristics among the four groups) and hereby suggests that to take into account multiple testing here, the critical value of p (kruskal-wallis with pairwise comparison) therefore should be 0.05/number of variables in the table, we respectfully disagree. We find it unusual to perform such corrections on baseline patient characteristics/descriptive data and consequently we have not done this but we are willing to do it on Editors' discretion.

Question 17

The question was "Tissues were immediately divided into smaller pieces of 50-100 mg, placed in separate tubes containing a chilled mitochondrial preservation buffer (BIOPS)". How did the authors manage to perform all the experiments with the very small sample obtained at the follow-up with the percutaneous liver needle biopsy?

The authors answered: "Thank you for the correction. In the manuscript we have now clarified further that the division of the tissue in the smaller pieces of 50-100 mg only took place at baseline (baseline biopsy). At 12 months the entire percutaneous sample was placed in buffer and subsequently transported to the laboratory to undergo HRR. The biopsies at 12 months were not divided as stated in the manuscript. This has been corrected".

The weight of the individual ultrasound-guided liver specimens is about 17 mg with larger needles, such as 14-gauge core.

Since part of the biopsy was used for histology, how did the authors manage to make the analyses of this smallest liver samples? Did they refine the technique? In fact, at baseline they used 50-100 mg of liver tissue.

Answer: Thank you. We have initially misunderstood your question.

As already stated in line 482 ('Study design and anthropometrics') we took out *three* ultrasonically guided TruCut (per biopsy: needle diameter 1.2 mm, length 10-15 mm, estimated volume of sample 15-25 mm²) liver biopsies at 12 months from each study subject. We used the same entry point in the right liver lobe in each patient; one biopsy for histology, one liver biopsy for respiratory analyses and the last biopsy was snap-frozen and biobanked (and used for mtDNA analysis).

At baseline we initially divided the liver tissue up into smaller pieces of 50-100 mg (as the tissue would be used for multiple purposes) but only 2-4 mg of this liver tissue was placed in the oxygraph (see 'Mitochondrial respirometry protocols' line 498)

Changes in MS: None

Reviewer #2 (Remarks to the Author):

Thank you for the responses you provided to my previous comments. I find them satisfactory. Furthermore, I appreciate the addition of the mitochondrial/nuclear DNA ratio after bariatric surgery, which strengthens your conclusion.

Thank you reviewer 2. We are pleased to hear that. Below see the corrections.

Here are additional suggested modifications from some of the added parts of the revised version of the manuscript:

1)Lines 144-149 : Reference 29 demonstrated that VAT display greater oxidative phosphorylation rate than SAT. Therefore, it is unclear why the authors refer to a lower respiratory reserve in VAT.

Answer: Thank you for your comment. Indeed, in ref 29 the mass specific (but not mtDNA corrected) oxidative phosphorylation is greater in VAT than in SAT.

However, in lines 144-149 we refer to the lower *respiratory reserve capacity* (not oxidative phosphorylation) found in VAT (fig 3e, ref 29) which is a 'measure' of how close the maximal oxidative respiration (GMOS) is to the uncoupled respiration (FCCP) which indicates that the VAT performs closer to its' maximal capacity than SAT. The reserve capacity is FCCP/GMOS.

Changes in MS: None

2)Line 382: remove 'but' at the end of the first sentence.

Answer: This has been corrected

3)Line 385: remove 'increase' before gene.

Answer: This has been corrected

4)Line 386: replace 'significantly' by 'significant'.

Answer: This has been corrected

5)Line 410: 'of' instead of 'og'.

Answer: This has been corrected

6)Line 411: remove 'In this case... mitochondria.'

Answer: This sentence has been deleted

7)Line 412: remove 'measure' before mitochondrial.

Answer: This has been corrected

8)Line 420: Not clear what the authors mean by 'The resolution is not near the resolution one can obtain with the ex vivo technique'. PET is by far the most sensitive technique there is, with a sensitivity in the range of pico to femtomolar. It may be more accurate to state that in vivo techniques usually provide proxy measures of mitochondrial function such as ATP/creatinine ratio or oxygen consumption.

Answer: Thank you for your comment. We here refer to NMR based techniques not PET techniques, however we have been unclear in our wording: '*For the liver, substrate metabolism can be measured in vivo by e.g. 31 phosphorous nuclear magnetic resonance, and oxygen consumption by dynamic oxygen 17 (17O) magnetic resonance imaging. But the resolution is not near the resolution one can*

obtain with the ex vivo technique has now been changed to ‘...dynamic oxygen 17 (¹⁷O) nuclear magnetic resonance spectroscopy (NMR)’.

This has been changed in the MS line 417

Reviewer #3 (Remarks to the Author):

The authors have addressed the issues I raised in the previous review.

Answer: Thank you, we are pleased to hear that.

However, the additional discussions regarding T2DM made me realize that I did not see any discussion of T2DM resolution in response to surgery. Did any of the participants see a resolution in their T2DM status, and if so, how many, and were there enough to see if any of the outcomes associated with resolution?

Answer: Thank you. This is a valid comment, but we chose to not include this in the discussion due to, as you mention yourself, too low a number for statistically testing the potential impact.

At baseline 15/62 study subjects were diagnosed with T2DM.

12 months postoperatively (study cohort n=21) 2/19 had T2DM. At baseline this number was 4/21, meaning that 50% had T2DM resolution in our follow-up cohort. Due to this small number we refrained from statistically testing if resolution of diabetes impacted change/increase in respiration.

Change in MS: None

Minor Issues

1. The wording of baseline differences should change from increased/decreased, to higher/lower

Answer: Thank you. We have changed the wording where relevant.

2. Line 335: “Secondly, did we not prove any correlation ..” should be “we did not”

Answer: This has been corrected

REVIEWERS' COMMENTS

Reviewer #1 (Remarks to the Author):

No further questions.

Reviewer #2 (Remarks to the Author):

Thank you for your response to my comments. I have no further issue.